# MINFLUX microscopy resolves subunits of the cardiac ryanodine receptor and its 3D orientation in cells

Alexander H. Clowsley [1,6], Anna Meletiou [1,6], Radoslav Janicek [1], Alexandre F. E. Bokhobza [1], Evelina Lučinskaitė [1], Gabriela Bleuer[1], Isabelle Jansen [2], Peter P. Jones[3], William E. Louch [4,5] & Christian Soeller [1] ✉

The cardiac ryanodine receptor (RyR2) constitutes the molecular basis of the process of calcium-induced calcium release where activation of RyR2s can be locally regenerative. Here, we present purely optical data of RyR2 distribution with sub-molecular resolution by applying 3D MINFLUX microscopy. Using single-domain antibodies and DNA-PAINT we determine the location of individual RyR2 subunits with high precision (~3 nm) and resolve the 3D orientations of RyR2s in-situ. We measured labeling efficiencies of ~50%, implying RyR2 tetramer detection probability approaching 95%. In HEK293 cells, RyR2 expression was dense, with some clusters containing several hundred RyR2s. Ventricular myocytes from mice contained large clusters containing many tens of close-packed RyR2s, resolving apparent discrepancies between electron microscopy and previous super-resolution microscopy data. The methodology developed here reveals the full 3D morphological complexity of RyR2 channels and is applicable to other multi-subunit complexes in a variety of cell types.

The large (~2.3 MDa) cardiac type 2 ryanodine receptor (RyR2) protein acts as a calcium channel and controls $Ca^{2+}$ release from intracellular stores. It is now well-established that the organization of RyR2s into adjacent groups of channels or clusters is an important modulator of $Ca^{2+}$ release in cardiac ventricular myocytes (VMs), and thus cardiac contractile function. Using conventional super-resolution imaging we and others have shown that RyR2s are organized into clusters of widely varying sizes (with an average size of ~10 RyR2s[1,2]), and that these form the molecular basis of $Ca^{2+}$ release[1,3]. The 3D arrangement of RyR2 tetramers within clusters has been shown using 3D electron microscopy to be highly variable[4,5], exhibiting both adjoining and oblique interactions[6] that may affect the gating of interacting RyR2s.

Optical super-resolution imaging of RyR2s has become an important technique to characterize the malleable relationship between subcellular RyR2 distribution at the nanoscale and local, as well as cell wide $Ca^{2+}$ release. For example, RyR2 dispersion, a nanoscale RyR2 rearrangement that includes fragmentation of RyR2 groupings into more numerous, smaller clusters, without any change in overall channel number, was first observed in myocytes from failing hearts using dSTORM super-resolution imaging[7]. Similarly, nanoscale ryanodine receptor configurations were correlated to calcium sparks in cardiomyocytes using live cell photoactivated localization microscopy (PALM)[8]. Similarly, functional $Ca^{2+}$ signals where correlated using in situ fixed myocytes imaged with DNA-PAINT[9]. Despite the strong impact that optical super-resolution imaging of RyR2s in ventricular myocytes has had to date, several critical issues remain. While the importance of the relative orientation of RyR2s has been suggested by previous EM studies[4,5], non-equivocal determination of RyR2

[1]Department of Physiology, University of Bern, Bern, Switzerland. [2]Abberior Instruments GmbH, Göttingen, Germany. [3]Department of Physiology, School of Biomedical Sciences and HeartOtago, University of Otago, Dunedin, New Zealand. [4]Institute for Experimental Medical Research, Oslo University Hospital and University of Oslo, Oslo, Norway. [5]K.G. Jebsen Centre for Cardiac Research, University of Oslo, Oslo, Norway. [6]These authors contributed equally: Alexander H. Clowsley, Anna Meletiou. ✉e-mail: christian.soeller@unibe.ch

orientations using optical super-resolution imaging has remained elusive. Similarly, there are some apparent discrepancies between data from optical super-resolution imaging of RyR2s and 3D EM studies. While EM studies suggest a fairly close packing of RyR2s in larger clusters[4,5], RyR2 clusters imaged with optical super-resolution methods seem more sparsely filled with receptors[2].

Recently, more advanced optical super-resolution imaging approaches have become available, that have demonstrated truly molecular resolution approaching the single digit nanometer scale in biological samples[10,11]. MINFLUX microscopy pairs that with full, near-isotropic 3D resolution using a single objective[12]. We therefore examined if MINFLUX 3D imaging of RyR2s would allow some of the issues mentioned above to be overcome.

Our results demonstrate that MINFLUX 3D imaging of RyR2s combined with a small marker approach can determine RyR2 3D orientations in intact cells. We show, to the best of our knowledge, the first fully isotropic 3D light microscopy data of RyR2s in HEK293 cells and mouse ventricular myocytes where localization precision is better than 3 nm in all directions. We demonstrate how to optimize labeling and detection in MINFLUX 3D imaging and achieve 50% effective labeling efficiency for GFP tagged subunits, implying a homo-tetramer RyR2-GFP detection probability approaching 95%. The high density of RyR2 observed in large clusters in mouse myocytes resolves the apparent discrepancy between super-resolution and electron tomography data as we show that densities of RyR2s in large clusters are comparable between modalities if we assume a subunit detection efficiency of ~50%. Given increasing evidence that the neighbor arrangement of RyR2s on a scale of ~10 nm affects the coordination of RyR2 openings and cluster excitability[13], MINFLUX 3D based imaging of RyR2s provides an important assay for mechanistic studies of pathophysiological changes in the control of calcium release in cardiac myocytes.

## Results

### 3D MINFLUX imaging of RyR2s in HEK293 cells

An important consequence of using MINFLUX microscopy for the 3D imaging of RyR2s becomes apparent in 3D views of the labeling distribution in HEK293 cells expressing RyR2s (the RyR2-GFP fusion protein RyR2$_{D4365-GFP}$ as previously characterized[14]), as shown in Fig. 1. To visualize RyR2s, using MINFLUX imaging, the cells were stained with single-domain antibodies (sdABs or nanobodies) against GFP that harbor DNA-PAINT docking strands to enable MINFLUX DNA-PAINT imaging[15]. To visualize locations of RyR2 subunits with high precision, repeated raw MINFLUX localizations from the same DNA-PAINT imager (identified by the same MINFLUX trace ID – parameter TID[15]) were combined[12,15]. Each location obtained in this way originated from a single imager attaching to a DNA-PAINT docking strand on an sdAB. This results in localizations with a median precision of better than 2 nm in all three-dimensions (see, Supplementary Fig. 1). Notably, the resulting 3D MINFLUX data has nearly isotropic resolution in all three dimensions, which has previously generally required more complex 4PI microscopy with opposing objectives[16].

As expected, the distribution pattern of RyR2 labeling reflects the tubular ER morphology (Fig. 1A, x-y view) in some areas, and its highly three-dimensional architecture is apparent in y-z views (Fig. 1B). We compared the appearance of the RyR2 MINFLUX data with data obtained with conventional DNA-PAINT wide-field super-resolution imaging with axial resolution improved by biplane imaging[17]. When compared at the slightly lower lateral spatial resolution available in wide-field DNA-PAINT, lateral x-y views look broadly similar in both modalities once the MINFLUX data is laterally slightly blurred, albeit with a very notable improvement in z-resolution in the 3D MINFLUX data (Supplementary Figs. 2 and 3). The importance of 3D information is emphasized by inspection of a small image region that in an x-y view shows a group of adjacent RyR2s (Fig. 1C). In a y-z view it becomes clear

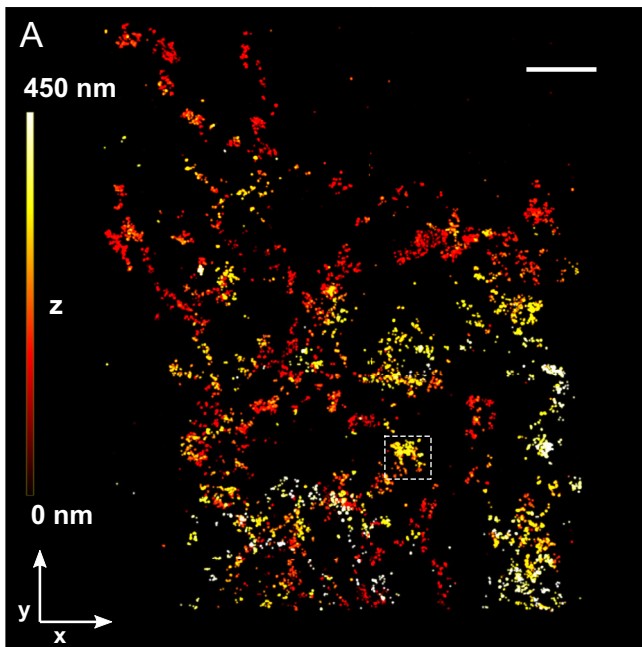

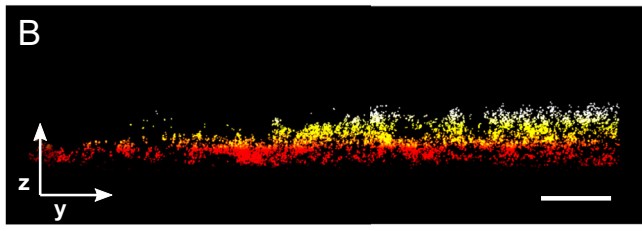

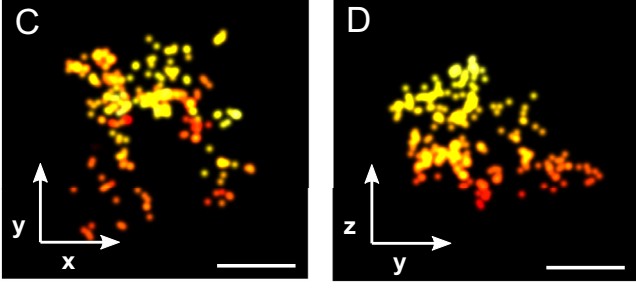

**Fig. 1 | 3D MINFLUX images of RyR2 sdAB labeling in HEK293 cells. A**. x-y overview of anti-GFP sdAB labeled RyR2s in a HEK293 cell stably expressing RyR2$_{D4365-GFP}$. **B**. The corresponding x-z view illustrates the complex 3D distribution of the localizations. **C**. A small region (dashed rectangle in A), shows that localizations that appear adjacent in an x-y view extend over considerable distances along the z-axis (**D**). Color indicates z-elevation. Scale bars **A**, **B**: 500 nm; **C**, **D**: 100 nm.

that the data has about equal extent in the axial direction (z, Fig. 1D) emphasizing the complex 3D architecture of the ER. The high 3D precision in the data is critical when determining cluster groupings of adjacent RyR2s as we show below.

### In situ measurement of effective labeling efficiency and detection probability with MINFLUX DNA-PAINT

In preliminary experiments with MINFLUX dSTORM photoswitching, we had noticed relatively sparse apparent labeling which was greatly improved by use of MINFLUX DNA-PAINT. We attribute this not to a better chemical labeling efficiency but more likely, in our hands, to a higher effective photo-detection probability of chemically bound markers with DNA-PAINT[18]. We implemented a quantitative in situ effective labeling efficiency assay to confirm our qualitative

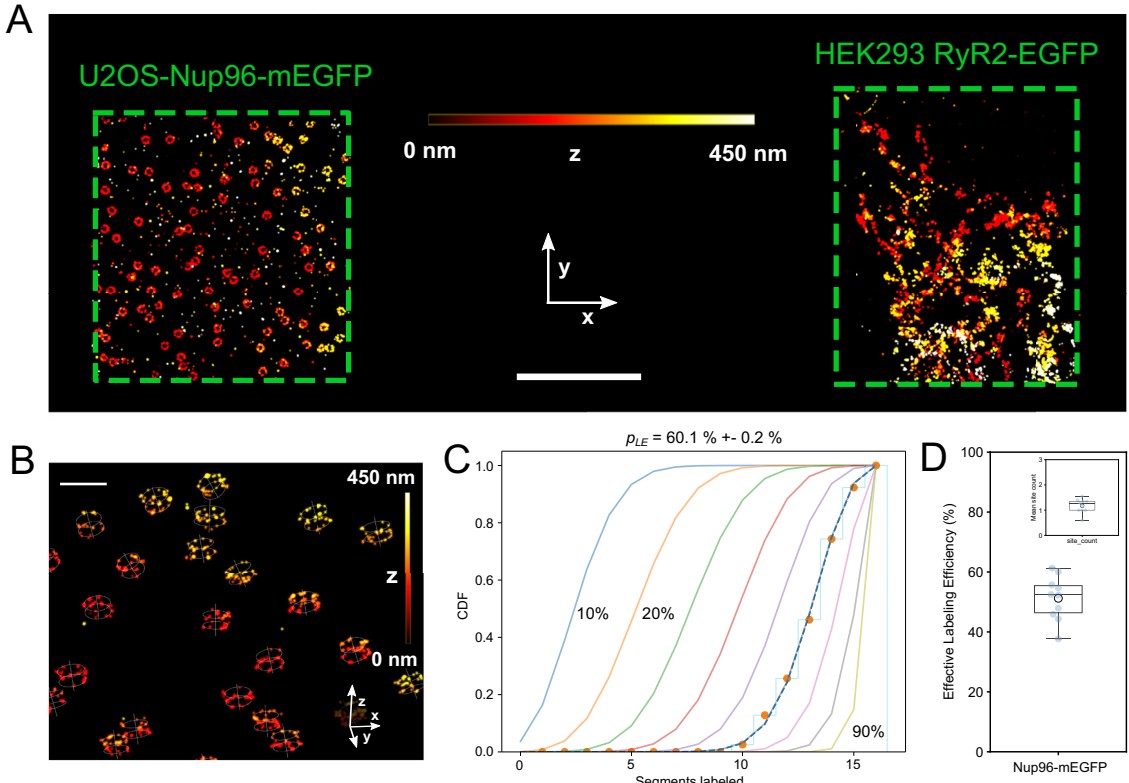

**Fig. 2 | Determination of effective labeling efficiency of anti-GFP sdABs in co-cultures of cells expressing RyR2$_{D4365-GFP}$ or U2OS-Nup96-mEGFP. A** MINFLUX 3D image of two adjacent ROIs (green boxes) that were imaged time multiplexed in one imaging run. The left ROI was selected in an U2OS-Nup96-mEGFP cell while the right one was located in a HEK293 cell stably expressing RyR2$_{D4365-GFP}$. Color indicates z-elevation. **B** 3D view of MINFLUX localizations in the U2OS-Nup96-mEGFP containing ROI with overlaid templates that identify locations and 3D orientation of NPCs. **C** Quantitative analysis of NPC data shown in B by overlaying the cumulative histograms of labeled NPC segments (0–16) with predicted curves for various values of $p_{LE}$ and line of best-fit (dashed) at $p_{LE} = 60.1 \pm 0.2$ %. **D** Across

experiments on average an effective labeling efficiency of 51.2% ± 2.3% was obtained ($N = 10$ MINFLUX data sets from $n = 3$ biological replicates). The inset shows the average number of site visits per U2OS-Nup96-mEGFP site, which is very close to 1 (1.18 ± 0.09, $N = 10$ MINFLUX data sets from $n = 3$ biological replicates). Box plots show the median (center line), the 25th and 75th percentiles (bounds of the box), and the mean (large circular marker). Whiskers extend to the most extreme data points within 1.5x the interquartile range, and all individual data points are displayed as a swarm overlay. Scale Bars **A**: 2 μm, **B**: 200 nm. Source data are provided with this paper.

observations as in MINFLUX DNA-PAINT care needs to be taken to fully collect localizations from the available marker distribution[18]. This aims at maximizing the marker detection probability P$_{detect}$ which can be conveniently monitored using the labeling of Nup96 proteins within the nuclear pore complex[18] (as recently established as a robust assay to measure effective labeling efficiency in conventional super-resolution imaging[19]). We therefore co-cultured U2OS-Nup96-mEGFP cells with HEK293 cells stably expressing RyR2$_{D4365-GFP}$ and then labeled these co-cultures with anti-GFP DNA-PAINT sdABs. We proceeded to record MINFLUX 3D images of Nup96-mEGFP labeling in U2OS-Nup96-mEGFP cells alongside labeling of RyR2-GFP in adjacent regions of interest (ROIs) in nearby HEK293 cells so that MINFLUX scanning multiplexed between these two ROIs, acquiring a dataset containing U2OS-Nup96-mEGFP 3D distribution adjacent to data of 3D staining of RyR2$_{D4365-GFP}$ (Fig. 2A, compare also Fig. 1). To estimate effective GFP labeling efficiency from the NPC 3D data we fitted a double ring template into imaged NPC structures to account for variations in 3D orientations of NPCs within the nuclear envelope (Fig. 2B). We used the detected and 3D aligned NPC localizations to measure the number of labeled segments (up to 16 in total with 8 on the cytoplasmic side and another 8 on the nucleoplasmic side). The obtained histograms were compared to the expected distributions for different labeling efficiencies to obtain an estimate of the effective GFP labeling efficiency $p_{LE}$. Further details of the template fitting procedure and the associated analysis are illustrated in Supplementary Fig. 4. For the NPC data shown in Fig. 2B this

approach yielded $p_{LE} = 60.1\% \pm 0.2\%$ (Fig. 2C). In addition, we analyzed the geometry of NPCs and obtained the expected ring spacing of ~50 nm (Supplementary Fig. 5). Consistent with the comparison between MINFLUX and SMLM biplane data of RyRs (Supplementary Fig. 3) MINFLUX based NPC analysis benefited from the high axial resolution in 3D MINFLUX imaging and provided more precise geometry parameter estimates as compared to widefield SMLM data (Supplementary Fig. 5C–E).

On average, in such combined RyR2$_{D4365-GFP}$ and U2OS-Nup96-mEGFP data sets, the effective GFP labeling efficiency $p_{LE}$ was 51.2% ± 2.3% ($N = 10$ MINFLUX data sets from $n = 3$ biological replicates, Fig. 2D). This confirmed that imaging durations were sufficient to fully capture the information from anti-GFP sdABs in the sample thus maximizing the marker detection probability P$_{detect}$[18]. This required extensive imaging durations (from 5 to 11 h) to ensure capturing all chemically labeled locations in the sample and helped ensure that information from labeled RyR2$_{D4365-GFP}$ was also fully captured, as the marker detection probability P$_{detect}$ should be about equal regardless of the specific target as it mainly depends on the DNA-PAINT attachment. In addition, the assay was used to measure the number of visits per site[18] which was 1.18 ± 0.09 ($N = 10$ MINFLUX data sets from $n = 3$ biological replicates, Fig. 2D inset) confirming that virtually no overcounting of the DNA-PAINT localizations is taking place which is useful for the interpretation of the observed labeling densities of RyR2s derived from the number of localizations per area or volume.

## Resolving RyR2 tetramers and RyR2 orientation in 3D

Having confirmed that we achieve substantial marker detection, we aimed to resolve the individual subunits making up the RyR2 homo-tetramer given the superior resolution available with MINFLUX 3D imaging. Among the marker systems available to visualize RyR2s with fluorescence super-resolution microscopy, the size of some marker combinations may defeat the purpose of achieving truly molecular resolution[20]. Labeling with primary and secondary antibody (AB) labeling systems while aiming for molecular imaging resolution is complicated by the considerable size of IgG molecules (7–12 nm, Fig. 3A). By contrast, the use of single-domain antibodies (sdABs) in conjunction with the relatively compact fluorescent proteins (FPs) should encounter smaller linkage errors, on the order of ~3–4 nm (Fig. 3A). Insertion of GFP after residue D4365 has been previously performed and the resulting RyR2$_{D4365-GFP}$ has been shown to be functionally essentially indistinguishable from wild type RyR2[14]. The locations of the 4 inserted GFPs in the RyR2 homotetramer were previously localized by single particle cryoEM and found to be arranged on the corners of a square with ~16 nm side length that is oriented at nearly 45 degree to the square outline of the tetramer itself[14] (Fig. 3B). Imaging HEK293 cells stably expressing RyR2$_{D4365-GFP}$ stained with an sdAB against GFP we could localize Atto-655 conjugated DNA-PAINT imagers transiently binding to these sdABs carrying complementary docking strands, with high precision in 3D, as shown in Figs. 1 & 2. Given the relatively small size of sdABs we aimed to localize individual subunits of a RyR2 tetramer.

We hypothesized that small groups of adjacent subunit localizations that we observed in MINFLUX data sets of RyR2$_{D4365-GFP}$ labeled with anti-GFP sdABs may represent adjacent subunits of the same RyR2$_{D4365-GFP}$ tetramer. Figure 3C shows an overview of a region of RyR2 labeling (color coded for the z coordinate) indicating a small region of interest that is shown magnified in Fig. 3D. Three areas are shown, magnified further, in Fig. 3E and slightly rotated in 3D when necessary, to display 4 (Fig. 3Ei) or 3 (Fig. 3Eii & iii) punctate foci of staining in the plane of rendering. These arrangements are suggestive of corresponding to adjacent subunits of a single RyR2. Candidate positions, given the geometrical arrangement of GFP residues in RyR2$_{D4365-GFP}$, are indicated in Fig. 3Fi-iii, with the individual RyR2 tetramers oriented in 3D as indicated in the axes shown in Fig. 3Ei-iii. This data shows that, when at least 3 subunits of a tetramer are labeled and detected this allows the 3D orientation of an RyR2 to be estimated with good precision within a cell. This is a unique ability of MINFLUX 3D imaging and demonstrates the potential to fully resolve the placement of multi-subunit complexes, such as RyR2, within cells and tissues.

Due to the relatively dense labeling detected in the MINFLUX DNA-PAINT datasets with optimized DNA-PAINT acquisition (Fig. 2) automatic detection of candidates for such directly labeled adjacent subunits proved difficult. Often, it was not possible to unequivocally assign subunit positions to individual candidate receptor positions in densely stained regions. In principle, such an automated assignment would enable estimating the effective labeling efficiency in situ from the fractions of receptors with 1 to maximally 4 labeled subunits and provide an alternative to the use of NPC based calibration.

## RyR2 clustering in HEK293 cells

Our estimate of effective GFP labeling efficiency of ~50% would imply a high likelihood to detect an RyR2 of ~95% (i.e. with at least one GFP containing subunit labeled), if EGFP accessibility in RyR2$_{D4365-GFP}$ is similar to that in U2OS-Nup96-mEGFP, see also the discussion. Prior to RyR2 clustering analysis we combined closely adjacent localizations (as these may have originated from the same sdAB due to repeated visits to a single DNA-PAINT docking strand, although the mean site-visit numbers should be close to 1, see also Fig. 2D) by DBSCAN clustering and combining such groups into a single event (using a distance

ε of 1 nm). We proceeded to analyze the three-dimensional distribution of the resulting subunit locations with the aim to achieve improved insights into the underlying RyR2 distribution.

To determine groupings of RyR2s that may be at a distance to effectively cross-talk via their released Ca$^{2+}$, we carried out clustering analysis to identify functionally coupled groupings of RyR2. Using a cutoff distance for the fire-diffuse-fire activation[21] of neighboring receptors via the mechanism of CICR by an open RyR2 (typically chosen around 100 nm based on Ca$^{2+}$ diffusion modelling[3,22]), DBSCAN clustering with ε of 100 nm provides a mechanistically motivated clustering approach to determine functionally coupled RyR2 clusters. As shown in the distribution of RyR2 in a HEK293 cell staining in Fig. 4A, with this signaling distance several clusters are very large spanning several microns. Here the three largest clusters contain >68% of all RyR2 labeling, each containing several thousand subunit locations. The high 3D resolution reveals the three-dimensional complexity of the architecture of these large clusters as shown in Fig. 4B–D which detail the large cluster in the bottom left corner of Fig. 4A. In x-z and y-z views one can clearly see an ER loop that spans >100 nm in the axial direction (see also Supplementary Movie 1 for animated 3D views). On average, clusters contain 30.9 ± 5.2 subunits ($N$ = 10 MINFLUX data sets from $n$ = 3 biological replicates); with a correction that uses U2OS-Nup96-mEGFP effective labeling efficiency as indicative of the RyR2$_{D4365-GFP}$ effective labeling efficiency (see discussion), clusters contain on average 58.8 ± 8.2 subunits (Fig. 4E). As illustrated in Fig. 4A most RyR2s are in large clusters, so that the three largest clusters together contain 34.8% of the total RyR2s on average (Fig. 4F, $N$ = 10 MINFLUX data sets from $n$ = 3 biological replicates). When corrected subunit numbers are converted to the number of RyR2s (by diving by 4), clusters contain a mean of 14.7 ± 2.1 RyR2s (Fig. 4G). This number seems moderate when noticing the very large clusters and can be explained as the mean cluster size is greatly affected by the presence of many small clusters, as seen in typical cluster size histograms (Supplementary Fig. 6). From a calcium signaling point of view the remarkably large functional groupings dominate, compatible with the idea that local spontaneous release from individual RyR2s can quickly travel over micron distances through the cell by spreading between adjacent receptors via the process of calcium-induced calcium release[23].

## MINFLUX imaging of RyR2s in ventricular myocytes

Using the recently characterized PA-RFP RyR2 knock-in mouse line[24] we also carried out MINFLUX imaging of RyR2s in isolated ventricular myocytes. Myocytes from PA-RFP RyR2 knock-in mice express RyR2$_{T1365-PATagRFP}$, an RyR2 with tagRFP insertion at the equivalent site to the previously structurally characterized RyR2$_{T1366-GFP}$[25]. For initial confocal imaging and to localize RyR2 clusters near the surface sarcolemma (Fig. 5A, B) we first stained with sdABs conjugated to Alexa 647. For molecular resolution 3D MINFLUX imaging, PA-RFP RyR2 mouse myocytes were instead labeled with anti-tagRFP sdABs that harbor DNA-PAINT docking strands as a small marker system. Using MINFLUX DNA-PAINT imaging, regions equivalent to those appearing blurred in confocal images (Fig. 5B) were clearly resolved into punctate groups of localizations (Fig. 5C). Using the subunit assignment approach, described above, followed by DBSCAN clustering with ε of 100 nm to capture functionally coupled clusters of RyR2s, the underlying RyR2 clusters were clearly revealed. Many of the prominent large clusters extend over distances >100 nm and contain several hundred subunits as shown by the color coding in Fig. 5C. The mean cluster size is, as in HEK293 cells above, strongly affected by the presence of many smaller clusters (which outnumber the prominent but fewer extended clusters) resulting in a mean cluster size of 9.0 ± 0.5 SUs or 16.5 ± 0.9 SUs when only looking at clusters with more than one SU ($N$ = 22 MINFLUX data sets from $n$ = 5 cell isolations, Fig. 5D). Isolated individual localizations are pointed out in Fig. 5E. Nevertheless, the majority

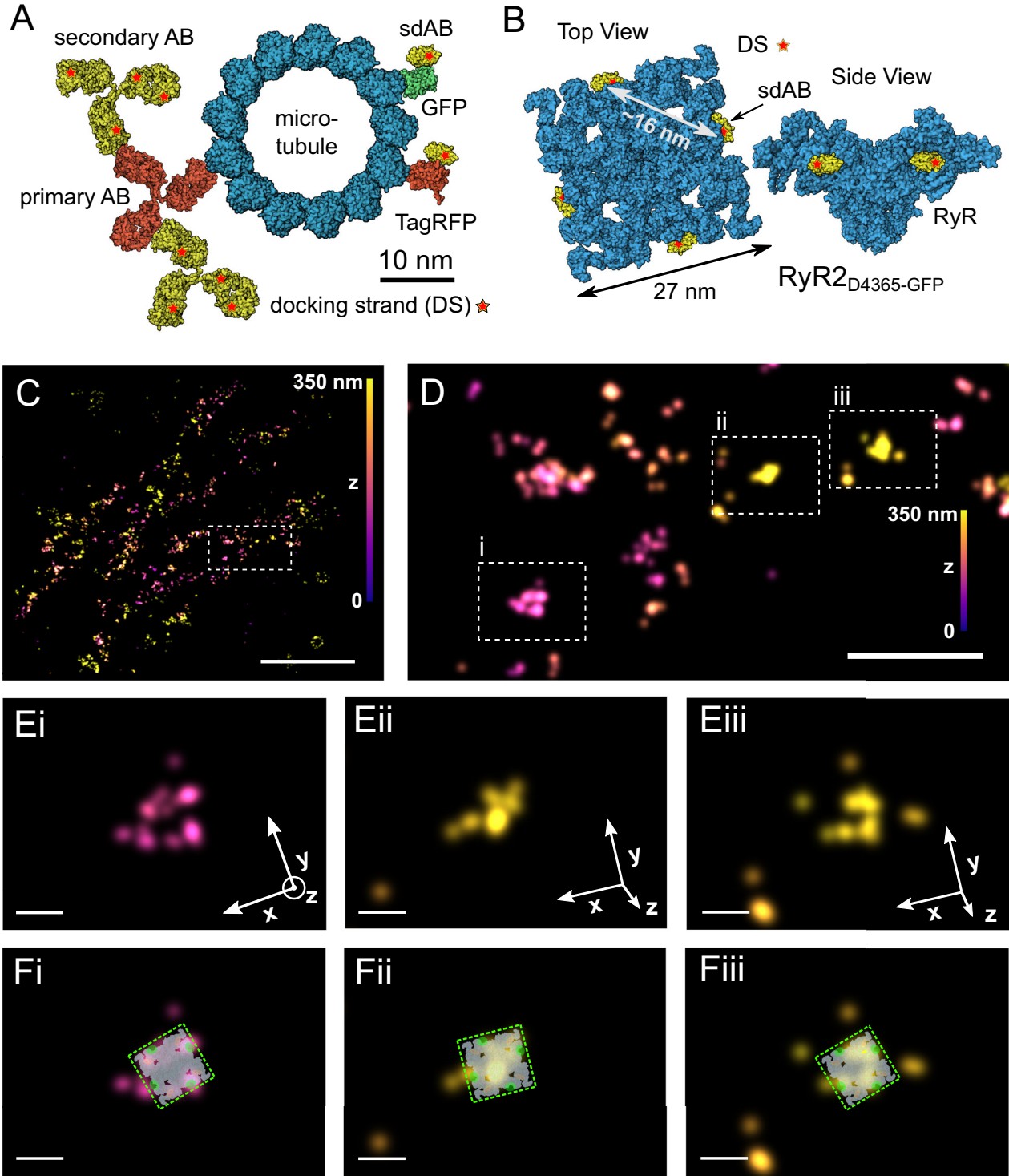

**Fig. 3 | Marker systems for RyR2 MINFLUX imaging and observation of tetramers. A** Schematic of relative target and marker sizes showing a primary/secondary AB labeling approach versus a GFP-sdAB or TagRFP-sdAB approach. A microtubule in cross-section illustrates target protein size (diameter ~25 nm) which is similar to the size of RyR2 (panel inspired by ref. 20). **B** Schematic of RyR2$_{D4365\text{-}GFP}$ with approximate locations of anti-GFP sdAB binding sites according to structural data from ref. 14. **C** Overview of a MINFLUX 3D data set of RyR2$_{D4365\text{-}GFP}$ labeling in a HEK293 cell indicating a smaller region of interest (ROI, white box) enlarged in (**D**). **D** 3 ROIs (i-iii) are indicated where the localization pattern is compatible with several subunits belonging to an RyR2 tetramer. **E** Enlarged view of ROIs in (**D**), rotated into the plane of the candidate RyR2 tetramer and also overlaid with the likely RyR2 orientation in (**F**). Color indicates z-elevation. Scale bars **C**: 1 μm, **D**: 200 nm, **E**, **F**: 20 nm.

of RyR2s reside in large clusters, where a cluster is classified as large if it contains at least 50 SUs, with on average 72.4 % ± 2.3 % of all RyR2s residing in large clusters ($N = 22$ MINFLUX data sets from $n = 5$ cell isolations, Fig. 5F).

The locations of the TagRFP insertion at T1365 map to a square with side length of ~20 nm that is approximately aligned with the four corners of the intact RyR2 tetramer[25] (see also Supplementary Fig. 7). In a detailed view of an RyR2 cluster shown in Fig. 5G, several subunit

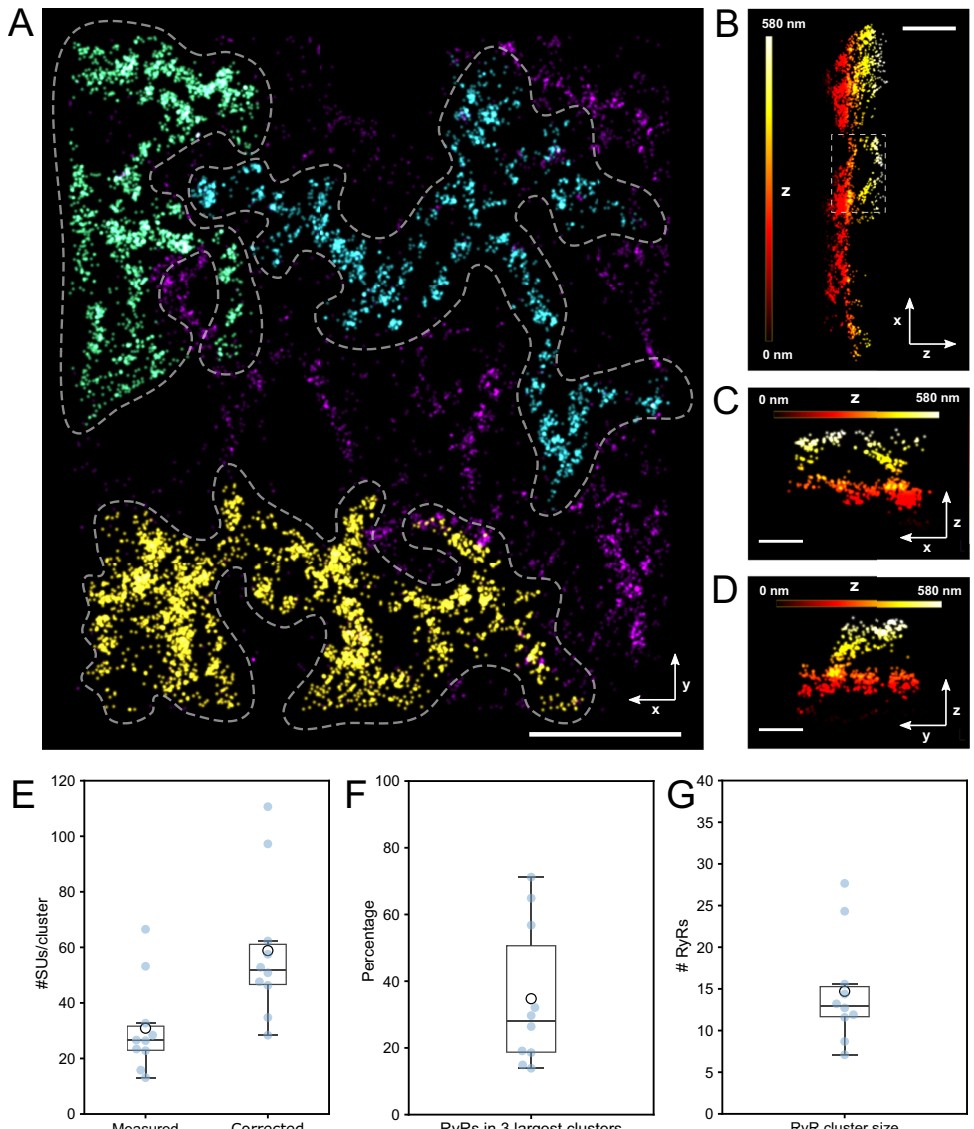

**Fig. 4 | Clustering of RyR2 subunit localizations in HEK293 cells. A** Overview of RyR2 subunit clustering in a HEK293 cell expressing RyR2_{D4365-GFP}. Clustering was determined with DBSCAN ($\varepsilon = 100$ nm) and using the 3D coordinates of subunit locations. Coloring indicates cluster size. **B** x-z view of the large yellow colored cluster in (**A**). **C**, **D**. x-z and y-z views of the loop-region indicated in B. Coloring in B-D indicates z elevation. **E** Boxplots of mean cluster sizes measured in SU/cluster as counted from data (Measured) and after correction with an indicative effective labeling efficiency (Corrected). $N = 10$ MINFLUX data sets from $n = 3$ biological replicates. **F** Fractions of RyRs present in the three largest cluster indicating that a

sizable fraction and frequently the majority of RyRs are within a few giant clusters. $N = 10$ MINFLUX data sets from $n = 3$ biological replicates. **G** Mean cluster sizes across all data sets obtained by using labeling efficiency corrected data from E and division by 4 to obtain RyR2 numbers from SU numbers. $N = 10$ MINFLUX data sets from $n = 3$ biological replicates. Box plots show the median (center line), the 25th and 75th percentiles (bounds of the box), and the mean (large circular marker). Whiskers extend to the most extreme data points within 1.5x the interquartile range, and all individual data points are displayed as a swarm overlay. Scale bars **A**: 1 μm; **B**: 500 nm; **C**, **D**: 250 nm. Source data are provided with this paper.

localizations are highlighted that are compatible with the position of individual tetramers as indicated in the receptor overlays. In several locations there are 3 labeled subunits; in two places all 4 subunits are labeled suggesting that MINFLUX can resolve the subunits of an RYR2 tetramer and thus directly show its 3D orientation within the cell.

Converting the number of subunits per cluster to RyR2 numbers (by dividing by 4) results in a mean RyR2 cluster size of $2.3 \pm 0.1$ RyR2s or $4.1 \pm 0.2$ RyR2s when including only clusters which contain more than one SU (Fig. 5Hi, $N = 22$ MINFLUX data sets from $n = 5$ cell isolations). These mean cluster sizes seem comparatively small when compared to the prominent large clusters that dominate the visual appearance (see Fig. 5C). This is largely explained by the fact that there are many more small clusters than large clusters, in fact large clusters (>50SUs, i.e. containing >12.5 RyR2s) make up less than 5% of all

clusters when counted on a per cluster basis ($4.7 \pm 0.3\%$, $N = 22$ MINFLUX data sets from $n = 5$ cell isolations, Fig. 5Hii) even though nearly three quarters of all RyR2s reside in such large clusters (Fig. 5F).

While the majority of the RyR2 clusters recorded near the cover slip reside in peripheral couplings with the surface sarcolemma, occasionally clusters extending into the myocyte are seen that are likely in internal couplings with transverse tubules that invaginate at the surface sarcolemma (Fig. 5Ii, ii). The clusters in peripheral couplings, while close to the surface sarcolemma, exhibit significant 3D structure that deviates from a purely flat 2D plane arrangement. These peripheral RyR2 clusters are often seen to subtly curve away from the coverslip surface which becomes apparent with the high axial precision available in MINFLUX 3D datasets (see Supplementary Fig. 8, see also Supplementary Movie 2 for animated 3D views).

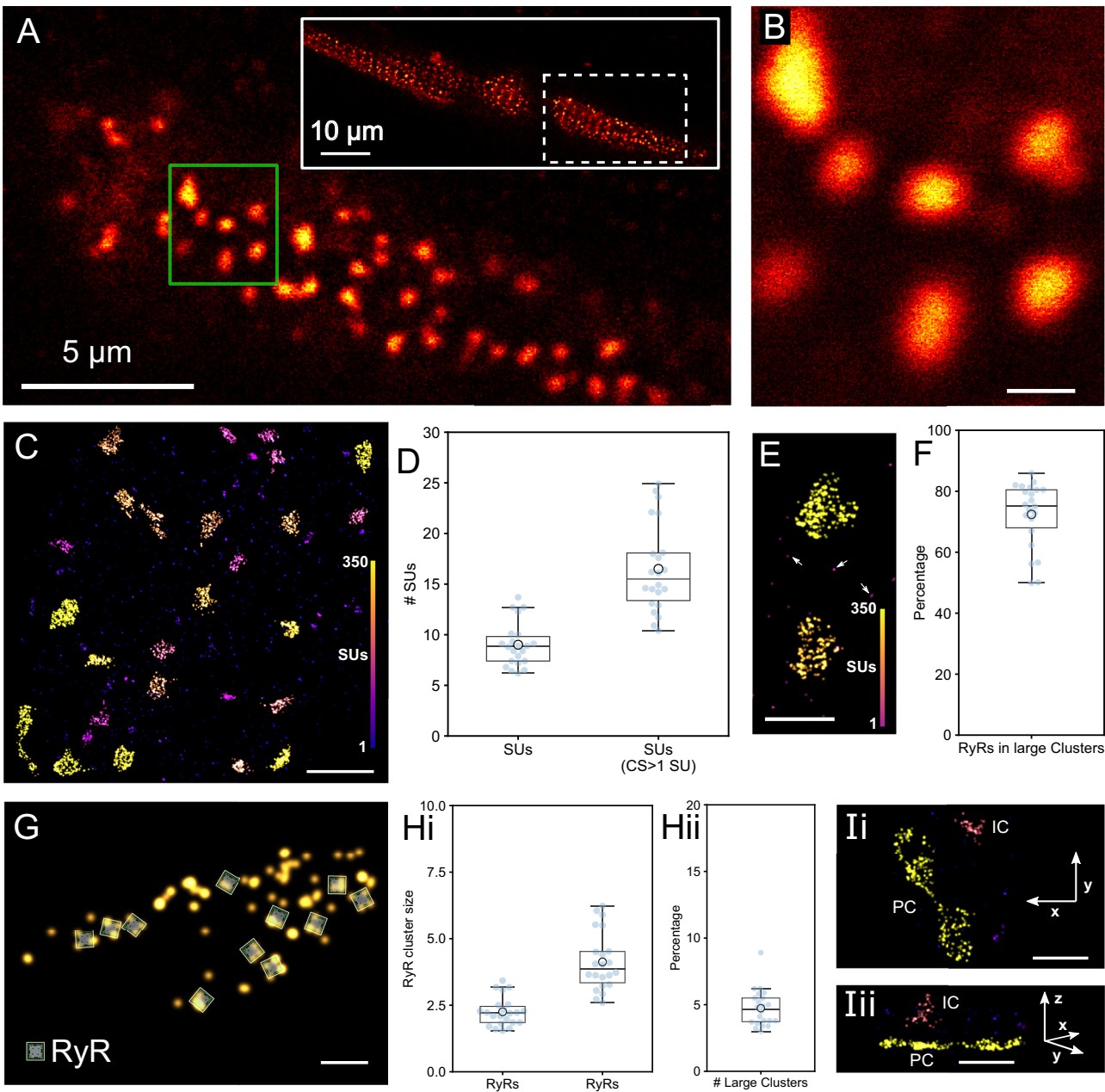

**Fig. 5 | MINFLUX imaging of RyR2s in myocytes from PA-RFP RyR2 knock-in mice. A** RyR2 distribution in an isolated myocyte from a PA-RFP RyR2 mouse, stained with an anti-tagFP sdAB-Alexa647 and imaged in the confocal mode of the MINFLUX microscope. $N > 5$ independent repeats with similar results. **B** Enlargement of a region at the surface sarcolemma (green rectangle in **A**). **C** A MINFLUX DNA-PAINT 3D image, localizations are colored by the size of clusters as number of clustered subunits (SU) from DBSCAN cluster analysis. **D** Statistical analysis yields a mean cluster size of $9.0 \pm 0.5$ SUs or $16.5 \pm 0.9$ SUs when only looking at clusters with more than one SU – labeled "CS > 1 SU" (N = 22 MINFLUX data sets from $n = 5$ cell isolations). **E** A detail view containing 2 large clusters and a number of isolated SUs (arrows). **F** The majority of RyR2s reside in large clusters containing ≥50 SUs with on average $72.4\,\% \pm 2.3\,\%$ of all RyR2s in these large clusters (N = 22 MINFLUX data sets from $n = 5$ cell isolations). **G** A large RyR2 cluster with a number of SUs in tretrad arrangements as indicated by RyR2-shaped overlays. **H** (i) Statistical analysis across datasets gives a mean RyR2 cluster size of $2.3 \pm 0.1$ RyR2s or $4.1 \pm 0.2$ RyR2s (CS > 1) when including only clusters which contain more than one SU. (ii) Fraction of large clusters (containing ≥50SUs, i.e. ≥12.5 RyR2s) with a mean of $4.7 \pm 0.3\%$ (N = 22 MINFLUX data sets from $n = 5$ cell isolations). **I** x-y (i) and x-z (ii) views of a peripheral coupling (PC) with the surface sarcolemma next to an internal coupling (IC) which is presumably arranged around a t-tubule invagination. Box plots show the median (center line), the 25th and 75th percentiles (bounds of the box), and the mean (large circular marker). Whiskers extend to the most extreme data points within 1.5x the interquartile range, and all individual data points are displayed as a swarm overlay. Scale bars **B**: 500 nm, **C**: 1 μm, **E**: 500 nm, **G**: 100 nm, **I**: 500 nm. Source data are provided with this paper.

## Relationship of MINFLUX data to previous EM tomography data

It is possible that the effective labeling of the tagRFP residues in RyR2$_{T1365\text{-}PATagRFP}$ is less than complete although we currently do not have access to an assay to measure it directly. As a proxy, we carried out a qualitative comparison between MINFLUX and EM tomography data of RyR2 clusters. We retraced EM tomography data from two previous studies[4,26] and used the obtained RyR2 cluster morphologies as the starting point to represent close to 100% detection efficiency of RyR2s (with the caveat that EM tomography may achieve this only for larger RyR2 clusters, small groups of RyR2s may be less sensitively

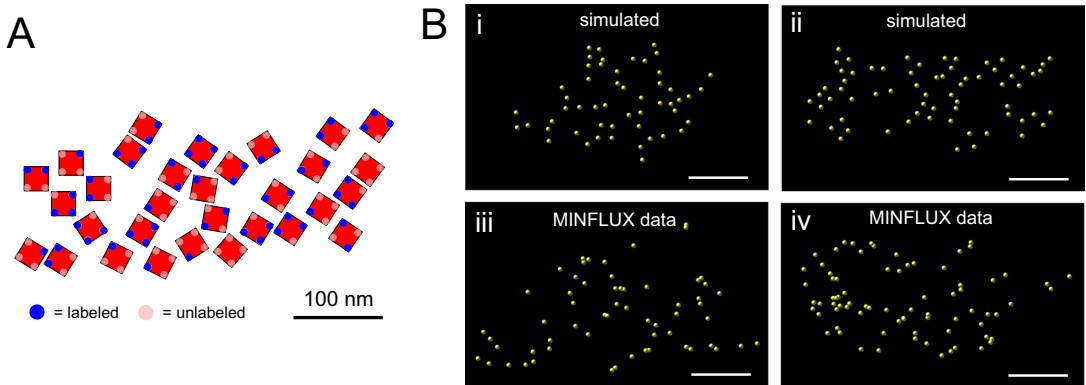

**Fig. 6 | Comparison of MINFLUX DNA-PAINT 3D RyR2 data with cluster morphologies from EM tomography data. A** RyR2 positions determined by EM tomography as traced from ref. 4, Fig. 1 therein, overlayed with the locations where sdABs can bind to TagRFP residues (see also Supplementary Fig. 7). From all possible binding positions labeled positions were selected randomly (shown as blue dots) with a probability $p_{SU} = 0.5$. **B** 3D localization errors ($\sigma = 3$ nm) were added to traced EM data from ref. 26, Fig. 2 therein (i) and[4], Fig. 1 therein (ii) to generate simulated subunit clusters which appear broadly similar to actual MINFLUX 3D subunit clusters (iii,iv). Scale bars **A**: 100 nm; **B**: 250 nm.

detected). The traced RyR2 cluster layouts were used to generate corresponding tagRFP locations (see also Supplementary Fig. 7) and then simulate a subunit detection efficiency of 50%. Subunit locations randomly selected according to this procedure (blue dots in Fig. 6A) were used to mimic MINFLUX subunit localizations by also adding 3D localization errors with a standard deviation of ~3 nm. The resulting simulated subunit clusters (Fig. 6Bi, ii) qualitatively resemble the larger clusters in our MINFLUX 3D RyR2 data recorded in myocytes (Fig. 6Biii, iv). This suggests an effective labeling efficiency approaching 50% is compatible with the MINFLUX DNA-PAINT data recorded in RyR2$_{T1365-PATagRFP}$ myocytes, broadly similar to that obtained for MINFLUX imaging of U2OS-Nup96-mEGFP with anti-GFP sdABs above.

To address the question if the observed non-complete labeling is purely a property of the sdAB labeling system used here, we carried out a comparison of labeling approaches using widefield DNA-PAINT super-resolution imaging. DNA-PAINT imaging previously achieved the highest resolution among conventional optical super-resolution microscopy methods. In previous studies, RyR2 positions were estimated from the location of punctate DNA-PAINT labeling in 2D rendered super-resolution images. We produced such images using (1) DNA-PAINT data of sdAB labeled PA RFP RyR2 mouse myocytes or (2) DNA-PAINT data of primary/secondary AB labeled RyR2s in mouse ventricular myocytes. When these images were analyzed using a blob finding algorithm, similar as done previously[2], the nearest neighbor distance distributions of blobs (previously identified as RyR2 positions) were comparable (Supplementary Fig. 9) with a median nearest neighbor distance between puncta of 36.8 versus 41.5 nm (ABs vs sdABs, Supplementary Fig. 9A, B) that was not significantly different ($p = 0.31$). This suggests that the sdABs used in our MINFLUX imaging experiments have per se a broadly comparable RyR2 labeling efficiency as other antibody labeling systems used previously. The amplification available in primary-secondary AB labeling (due to several docking sites per secondary AB and possible multiple binding) should have a negligible effect on puncta distances, since due to the lower DNA-PAINT resolution (typically ~24 nm Fourier Ring Correlation (FRC) resolution) these should all be merged into the same punctum. We also rendered MINFLUX myocyte data into 2D super-resolution images and analyzed these with the same blob finding approach. In these MINFLUX 2D super-resolution images the median nearest neighbor distance between puncta was reduced to 21.8 nm (Supplementary Fig. 9C, significantly smaller than distances from widefield DNA-PAINT imaging both with ABs and sdABs, $p < 0.001$, respectively). This reduced mean distance was in turn comparable to the NN distance distribution generated from EM tomography data and simulating a subunit detection efficiency of 50% which resulted in a mean NN distance of 19.4 nm (Supplementary Fig. 9D). Both the mean and the shape of this simulated distribution resembled that of the distribution obtained from MINFLUX DNA-PAINT data.

## Discussion

The isotropic 3D resolution approaching the single digit nanometer scale achievable with MINFLUX microscopy is a significant advance over previous optical super-resolution techniques that opens the door to truly molecular scale imaging in the heart. The higher resolution available with MINFLUX prompted us to re-evaluate the labeling, imaging and analysis processes involved in fluorescence imaging of RyR2s in HEK293 cells and adult mouse ventricular myocytes. Our MINFLUX data supports a revised interpretation of the relationship between prior RyR2 data from EM tomography and super-resolution fluorescence microscopy that resolves some previously noted apparent discrepancies.

The use of a small marker system was critical to enable the resolution of subunit locations. The relatively large size of RyR2 tetramers in conjunction with the sufficiently small linkage error and high MINFLUX localization precision are expected to keep the total error (in 3D) below about 4 nm. Given the known location of GFP insertions which had been previously determined using cryo-EM[14,25] we show that estimating not only the precise location but also the 3D orientation of RyR2 tetramers in intact cells becomes feasible. Retrieving orientation information for RyR2 tetramers in situ is an exciting prospect of MINFLUX and related molecular imaging techniques that should also be applicable to other large protein complexes.

To reach the full potential of nanometer resolution fluorescence imaging the utility of a precisely known labeling location is essential. This is achievable using molecular tagging as with the fluorescent protein tags employed here and could exploit even smaller tags such as SNAP, Halo[27] and ALFA tags[28]. Insertion of these tags is now readily achievable and has the additional advantage that well characterized markers can be employed. A potential complication in practice may be the insertion into adult cardiac myocytes which generally requires viral delivery and for physiologically most relevant results will often involve generating a suitable genetically modified organism[8,29].

Similar to previous reports[30], we observed an apparently quite small effective labeling efficiency when we utilized dSTORM based dye

blinking in our preliminary MINFLUX experiments. We hypothesized that alongside photobleaching this resulted from stringent photo-detection requirements in MINFLUX imaging, in addition to the purely chemical labeling ($P_{chem}$) efficiency of labeling with the sdABs. The $P_{chem}$ of a sdAb is not expected to significantly differ between markers harboring either a photoswitchable dye or a DNA-PAINT docking strand. However, the photo-detection requirement should be easier to fulfil with MINFLUX DNA-PAINT imaging since fresh imager molecules can diffuse in and bind to docking strands on sdABs. We implemented a co-culture approach to enable measurement of labeling efficiency in situ using the robust NPC assay originally established by Thevatasan et al[19]. For MINFLUX we extended this to full 3D analysis using template fitting[31] for precise geometrical localization and analysis. Our data demonstrates that when care is taken to collect the data over an extended acquisition time frame (several hours per dataset) MINFLUX DNA-PAINT imaging can routinely achieve effective labeling efficiencies >50% for U2OS-Nup96-mEGFP (we measure the product of chemical labeling $P_{chem}$ and photo-detection efficiency $P_{detect}$[18]), similar to effective labeling efficiencies measured in conventional widefield super-resolution experiments[19]. This is also consistent with the qualitatively similar appearance of MINFLUX and widefield DNA-PAINT RyR2 labeling when compared in 2D and at the lower resolution available with widefield super-resolution imaging (Supplementary Fig. 3).

The NPC assay performed simultaneously with the RyR2 data acquisitions helps ensure that the detection probability $P_{detect}$ is high[18]. We have recently shown that the number of visits per site is on the order of one for the conventional DNA-PAINT markers used here due to site-loss associated with the extended MINFLUX scan duration making overcounting sites unlikely[18]. We explicitly confirmed this with the NPC assay in the experiments reported here (site counts shown in Fig. 2D). The chemical component $P_{chem}$ of the effective labeling efficiency of RyR2$_{D4365-GFP}$ could differ from that of U2OS-Nup96-mEGFP if the marker accessibility was quite different. The main case to discuss is the possibility if it were considerably lower (as equal or higher values are advantageous for data interpretation). This seems unlikely with the high density of RyR2 signal that was recorded, see Fig. 4 and the presence of many 3 or 4 member tetramer arrangements. An indirect support is also lent by the observation that the observed RyR2 density in clusters within cardiac myocytes is compatible with EM tomography data when an effective labeling efficiency of 50% is assumed, using a different sdAB (Fig. 6).

If the efficiency was comparable to U2OS-Nup96-mEGFP, then an average >50% likelihood of detecting a GFP containing RyR2 subunit predicts a ~94% probability of detecting an RyR2 by labeling at least one of its four subunits. In principle, the characteristic configuration of 4 subunits on RyR2s should also enable a direct way of estimating the effective labeling efficiency but it proved difficult to unequivocally assign subunit positions to candidate receptor positions in the densely stained samples. A template fitting approach similar to the NPC assay is desirable but complicated by identifying candidate subunit groupings to fit to. Fitting to all possible groupings is currently prohibitively CPU intensive due to the global minimization essential in template fitting. An alternative statistical approach or accelerating analysis with a suitable machine-learning based method may circumvent this and also eventually allow extension to more heterogenous multi-unit complexes.

Imaging of the NPC has become an important assay to quantify labeling efficiency[19] and has contributed to revealing the 3D architecture of the NPC in situ. By combining cryo-electron tomography with mass spectrometry, biochemical analysis, perturbation experiments and structural modelling a comprehensive architectural model of the human nuclear pore complex has been previously generated[32]. From this model, the spacing between cytosolic and nucleosolic rings of Nup96 protein locations was obtained as ~57 nm[33]. Super-resolution

imaging has since then been used to estimate these dimensions from fluorescence data with widefield SMLM based estimates of Nup96 ring spacing of 49.8[19], 50.2[31] and 51.2 nm[34]. The first MINFLUX measurement of this distance determined 46 nm[10] and a recent live cell MINFLUX study estimated a ring spacing of 51.5 nm[35]. In our experiments we measured a ring spacing of 49.6 nm (Supplementary Fig. 5), similar to previous super-resolution based measurements. A recent preprint reanalyzed published datasets and determined ring spacings closer to the predicted distance from the von Appen cryo-EM study[32], namely 57.5 nm, and a comparatively large value of 63.2 nm (from a 4Pi-STORM dataset)[36].

The approximately 10% difference between most reported super-resolution and the original cryoEM measures may be explained by (a) the finding that NPCs exhibit substantial plasticity and that the purification of the nuclear envelope as employed in prior cryoEM studies may influence the human NPC structure[37], (b) that the reported ring spacing in super-resolution studies sensitively depends on the z-axis correction that is applied to correct for refractive index mismatch[10], and (c) that some offsets may result from the labeling system employed. In accordance with the frequently reported ring spacing of ~50 nm in super-resolution studies, recent cryo-EM data of NPCs in the native cellular context yielded a reduced NPC height (by ~10%) compared with previous models that used data from purified samples[37]. Both in a previous MINFLUX study and our study an sdAB labeling system against GFP in Nup96−mEGFP was employed, with an up to 6−7 nm distance between the dye and the NUP96 attachment point[34]. Given these sources of differences between studies all reported values agree comparatively well. For the purposes of this study the focus has been on the ability to quantify effective GFP labeling efficiencies. With the maximum-likelihood fitting approach employed here labeling efficiency estimates are robust against small (~10%) geometry variations.

A remarkable property of MINFLUX is the essentially isotropic 3D resolution when operated with a 3D doughnut[10] (see also our Fourier Shell Correlation Resolution (FSC) measurements in Supplementary Fig. 1 which confirm near isotropic resolution). This may be rivalled by existing 4PI SMLM approaches[16,38] but at the cost of the complexity of working with opposing objectives. The single objective design of 3D MINFLUX microscopy markedly simplifies the preparation and mounting of biological samples, a factor which in our experience is difficult to overestimate. The high, near isotropic 3D resolution enabled the quantification of the intrinsic three-dimensional nature of RyR2 clusters which was previously difficult to quantify using other optical super-resolution imaging techniques. This allows clearly distinguishing the peripheral from internal couplings (Fig. 5I) and revealed the curved nature of peripheral couplings (Supplementary Fig. 8).

It is important to keep in mind that in fluorescence microscopy we only see dyes on markers, or in the case of DNA-PAINT, where an imager strand attaches to a docking strand on a marker, rather than the actual location of the protein targets themselves. In the scenario that (1) a substantial fraction of the target sites are occupied by markers with high specificity, (2) most or all of these markers are detected and precisely localized by the imaging system and (3) the resolved spatial detail is comparable (or not much smaller) than the marker size we can treat the obtained images as close replicas of the actual target site distribution. With the 3D resolution of MINFLUX, the high specificity and comparatively small size of anti-GFP sdABs, the large size of the proteins involved and the focus on maximizing detection efficiency as described here the image data should provide an accurate reflection of the underlying RyR2 distributions in 3D.

We used previous data from EM tomography studies[4,26] to simulate how cluster layouts obtained from EM tomography would appear in MINFLUX 3D imaging with ~50% subunit labeling efficiency. The appearance of the simulated subunit location data was visually similar

to the actual MINFLUX data we obtained. This observation provides a resolution to apparent discrepancies between RyR2 distribution patterns from previous super-resolution and EM data, where EM data from tomograms appeared to indicate a more dense packing of RyR2s[4,26] than observed in super-resolution imaging studies[2]. According to our data this can be largely resolved by the improved resolution of MINFLUX 3D imaging if care is taken to maximize what we term marker detection probability $P_{detect}$ to achieve the limits set by chemical labeling efficiency $P_{chem}$. The latter point is a critical additional requirement since in MINFLUX DNA-PAINT imaging one needs to acquire for sufficiently long durations which can be monitored with an NPC assay[18] as we use here. This interpretation is also supported by the puncta analysis in the different modalities (Supplementary Fig. 9) which showed qualitative agreement between puncta distances in MINFLUX and data simulated from EM tomography data. In conjunction with the in situ estimate of MINFLUX 3D labeling efficiency this approach provides a way to use the complementary data provided by fluorescence and EM modalities to help reconstruct true underlying molecular distributions. The MINFLUX based methods we introduce here provide advantages for applications in cardiac molecular imaging because of the comparatively simpler sample preparation, potentially faster imaging and data analysis, the ability to extend this to proteins not easily discerned in EM data and, at least in principle, compatibility with live cell imaging.

Previous data on RyR2 clusters in peripheral couplings, albeit in rat ventricular myocytes, provided a mean of 8.8 RyR2s/cluster estimated with 2D DNA-PAINT[2]. The values from 2D imaging may not be directly comparable to that from MINFLUX 3D imaging and the cluster algorithm was quite different while labeling efficiency was not measured. The raw subunit data with 9.0 SUs/cluster seem on first sight broadly similar, however, when converting to RyR2s by simply dividing by 4 this number reduces to 2.3 RyR2s/cluster. Accounting for likely incomplete labeling (we argue above ~50% is a reasonable estimate) and rounding all numbers of RyR2s for each cluster to whole numbers a mean size of ~5 RyR2s/cluster would result. Only considering clusters with at least 2 SU detections (4.1 RyR2s/cluster) and additionally considering an estimated labeling efficiency of ~50% we arrive at ~8.2 RyR2s/cluster. The biggest issues in comparing mean numbers across different modalities are likely (a) an increased sensitivity of MINFLUX 3D imaging to detect isolated signals and more generally (b) the reduction of the long tail distribution of cluster sizes to a single number (which is heavily distorted by a large number of isolated single/few SU clusters).

A key goal in grouping RyR2s into clusters is to identify groups of receptors that activate in unison to give rise to Ca²⁺ sparks and similar local release events. Due to calcium-induced calcium release (CICR) these functional groups are generally formed by RyR2s in close proximity and thus approaches to group RyR2s into clusters according to a distance criterion have been widely used. While the DBSCAN algorithm may not be optimal for detecting clustering in some biological scenarios[39], for CICR it accurately reflects the mechanism of fire-diffuse-fire activation of neighboring RyR2s and was therefore used with a 100 nm distance to identify groupings of receptors likely to activate in unison[3,22,40]. Analysis for other signaling distances is possible by adjusting the clustering parameter ε. Indeed, with this approach we identified very extended RyR2 clusters in the HEK293 model employed here. Similarly, large RyR2 clusters dominate the visual appearance of RyR2 distribution in ventricular myocytes (containing typically many tens to >100 RyR2s). From a Ca²⁺ signaling point of view we would expect these clusters to dominate the release during activation and also generate the majority of the spontaneous Ca²⁺ sparks observed in otherwise quiescent myocytes, a major component of the functionally important SR leak in these cells[41].

A significant advance of MINFLUX 3D imaging compared to previous super resolution modalities is the ability to identify the relative placement and orientation of single RyR2 channels within a cluster. Whilst cluster size was previously used to define the activity of clusters it is clear from more recent EM data that things are more complex[4], as (patho)physiological stimuli can alter the nanometer scale packing of channels within the cluster. Mathematical modeling has indicated that these changes critically alter both the fidelity and time course of spark generation[5,7]. Compared to EM based 3D methods, sample preparation for MINFLUX imaging is comparatively straightforward. It can routinely achieve single channel localization and orientation to enable probing of the plastic geometry of channels within a cluster, under a range of conditions, with good statistical coverage. Therefore, MINFLUX 3D imaging will be crucial to expand our knowledge of how intra-cluster geometry influences cluster function. Looking ahead, it is capable of multi-channel imaging and will also allow for simultaneous imaging of local cellular compartment architecture. In conjunction with such molecular resolution data and mathematical modelling as well as correlative functional studies[8,9,42], MINFLUX 3D imaging lays out a clear path to arrive at more precise and functionally relevant criteria to determine RyR2 functional groupings and refine our understanding of the RyR2 cluster concept for cardiac Ca²⁺ signaling.

We conclude that MINFLUX 3D imaging provides a powerful way to obtain molecular scale data on RyR2 distribution in cardiac cells with isotropic nanometer resolution. The additional information available from 3D MINFLUX imaging of RyR2s will be useful for the construction of more detailed mathematical models and improve our understanding of Ca²⁺ signaling in the cardiac dyad. In addition, 3D MINFLUX imaging of RyR2 distribution within cells constitutes a more sensitive assay of nanoscale structure of cardiac myocytes than previously available super-resolution methods and should be used to refine observations of alterations in cardiac patho-physiology[5,7].

Given the multi-channel imaging capability of 3D MINFLUX imaging there is great scope for future studies that combine 3D distribution information from other Ca²⁺-handling proteins and structures, such as transverse tubules and SR shape markers as well as other ion channels while maintaining high labeling efficiency.

## Methods

For experiments carried out at the University of Oslo, all animal protocols were performed in accordance with the Norwegian Animal Welfare Act and NIH Guidelines (NIH publication No. 85-23, revised 2011) and were approved by the Norwegian Food Safety Authority (permit number 8951). For experiments carried out at the University of Bern, all animal protocols were performed in accordance with the Swiss Animal Welfare Act and were approved by the Amt für Veterinärwesen of the Canton of Bern (permit number 35279). Housing and breeding were carried out in a departmental animal facility with controlled environmental conditions (22 °C, 40 % relative humidity, 12 h light/dark cycle), with mice having free access to laboratory chow and water. In this study we used adult male mice ($N = 5$, 2 – 12 months old).

### Preparation of HEK-293 cells stably expressing RyR2$_{D4365\text{-}GFP}$ for MINFLUX imaging

HEK293 cells stably expressing inducible RyR2$_{D4365\text{-}GFP}$ created in the laboratory of Dr. Wayne S.R. Chen[14] and maintained in the laboratory of Pete Jones, University of Otago, were grown as described further below. Genotyping for HEK-293 was carried out by Pete Jones at the University of Otago, and its GFP-specific expression pattern following induction was verified in the laboratory of Dr. Christian Soeller. They were grown at 37 °C with 5% $CO_2$ in Dulbecco's Modified Eagle's Medium – high glucose (Sigma), supplemented with 5% fetal bovine serum (Gibco) and 1% Antibiotic-Antimycotic (Gibco).

HEK293 RyR2$_{D4365\text{-}GFP}$ cells were seeded on precleaned coverslips treated with 0.1% Poly-l-lysine for 10 min. After 1–2 days, the expression of RyR2$_{D4365\text{-}GFP}$ was induced by 1 µg/ml doxycycline. 24 h after the induction, cells were fixed with 4% PFA in PEM buffer[43] (80 mM PIPES,

5 mM EGTA, 2 mM MgCl$_2$, pH 6.8) with 4% sucrose at RT for 10 min before being washed 3x with PBS. Coverslips with fixed cells were mounted on custom made slide chambers made from acrylic[44]. Cells were permeabilized for 10 min in PBS containing 0.1% Triton and then subsequently blocked using MP antibody incubation solution (Massive Photonics, München) for 1 h at RT.

For labeling of GFP residues on RyR2$_{D4365-GFP}$ for MINFLUX, cells were incubated with a 1:200 dilution of TAG-X2 anti-GFP (sdAB, clones1H1 & 1B2, batch N°3302002, Massive Photonics, München) harboring a conventional docking strand (MP DS3, undisclosed sequence, Massive Photonics, München) for 1 h at RT in MP antibody incubation solution. Samples were washed in DNA-PAINT buffer (PBS containing an additional 500 mM NaCl) 3x. These were imaged with a nominal 1 nM concentration of Imager 3 diluted in DNA-PAINT buffer.

For comparison widefield DNA-PAINT imaging in 3D biplane mode HEK293 RyR2$_{D4365-GFP}$ cells were labeled with TAG-X2 anti-GFP sdAb harboring a fast docking strand (MP FAST DS3, undisclosed docking sequence, Massive Photonics, München) and imaged using 0.5 nM Imager F3 ATTO 655 diluted in DNA-PAINT buffer with a camera integration time of 25–50 ms.

### Preparation of co-cultures of HEK-293 cells stably expressing RyR2$_{D4365-GFP}$ with U2OS-Nup96-mEGFP cells

HEK293 cells stably expressing inducible RyR2$_{D4365-GFP}$[14] and U-2 OS-Nup96-mEGFP (Cytion, clone195, # 300174) were cultured at 37 °C with 5% CO$_2$. U-2 OS-Nup96-mEGFP were not further authenticated after purchase, except for verification of the expected GFP expression pattern at the nuclear pore. HEK293 cells were maintained in Dulbecco's Modified Eagle's Medium – high glucose, GlutaMAX™ Supplement, pyruvate (Sigma), and U-2 OS cells were cultured in McCoy's 5 A medium (Fisher Scientific). Media were supplemented with 10% fetal bovine serum (Gibco) and 1% Antibiotic-Antimycotic (Gibco). For co-culturing, $4 \times 10^4$ HEK293 cells were seeded on pre-cleaned coverslips. After 2 days, the medium was refreshed and RyR2$_{D4365-GFP}$ expression was induced by the addition of 1 µg/ml doxycycline. 24 h after the induction, $1.2 \times 10^5$ U-2 OS were added in McCoy's 5 A medium. Co-cultured cells were fixed one day later and labeled in the same manner described above for single culture HEK293 MINFLUX experiments.

### Cardiac myocyte isolation from PA-RFP RyR2 mice

Genetically modified PA-RFP RyR2 mice on the C57BL/6/6 N background that harbor PA-TagRFP inserted after T1365, within exon 31 of the mRyR2 gene on chromosome 13 as described[8], were housed at the University of Bern and used to provide isolated myocytes. Single cardiomyocytes were isolated as previously described[45]. Briefly, the adult male mice were first anesthetized by intraperitoneal injection of pentobarbital (150 mg/kg of body weight). After the disappearance of the tail pinch reflex, the heart was rapidly excised, cannulated, and retrogradely perfused on a Langendorff system at 37 °C. Isolation buffer (Ca$^{2+}$-free modified Tyrode solution) containing 140 mM NaCl, 5.4 mM KCl, 1.1 mM MgCl$_2$, 1 mM NaH$_2$PO$_4$, 10 mM HEPES and 10 mM D-glucose (pH adjusted to 7.4 using NaOH) was first used to flush the heart of blood. Once fluids ran clear the perfusion was switched to an isolation buffer which included collagenase type II (160 U/mL, Worthington Biochemical Corporation, LS004177) and protease type XIV (0.21 U/mL, Sigma, P5147) for ~15 min. After digestion the heart was cut down into isolation buffer and the ventricles were diced into small chunks. The cell suspension was then filtered through a filter mesh (pluriStrainer PET 100 µm), the cardiomyocytes were sedimented and the Ca$^{2+}$ concentration gradually increased to 0.25 mM. Chamber slides with an attached coverslip were pre-treated with laminin (Gibco, 23017015) diluted in PBS 3:200. The chambers were left at room temperature overnight and subsequently washed with Ca$^{2+}$-free Tyrode's solution on the day of the isolation. Cell solution was pipetted onto these prepared chambers and allowed to settle for 2 h at 37 °C.

Subsequently, cells were fixed using 4 % PFA diluted from stock (Electron Microscopy Sciences, 15714) for 10 min at room temperature. The samples were washed with PBS for 10 min and then stored in a PBS solution containing 0.5 % BSA and 0.1 % sodium azide until later use.

### Labeling of myocytes from PA-RFP RyR2 mice for MINFLUX and widefield DNA-PAINT imaging

Fixed cardiomyocytes were permeabilized with 0.1% Triton X-100 for 10 min and then blocked with MP antibody incubation solution for 1 h. Cells were then incubated with Massive-Tag-Q anti-TagFP single-domain antibody (Clone 1H7, Massive-Photonics, München) harboring strand DS3 for 2 h at room temperature diluted 1:200 in MP antibody incubation solution. Samples were washed >3 times in DNA-PAINT buffer. For confocal imaging the cells were incubated with FluoTag® X2 anti-TagFP (#N0502, Clone 1H7, NanoTag Biotechnologies, Göttingen) diluted 1:50 in incubation solution made from PBS containing 2% BSA, 2% NGS, 0.05% Triton X-100 and 0.05% NaN$_3$ for 2 h at RT.

For comparison DNA-PAINT experiments, involving primary and secondary antibodies, after the samples were permeabilized the cells were blocked in PBS containing 1% BSA for 1 h at RT. Cell were then incubated with a ryanodine receptor monoclonal antibody (C3-33) (MA3-916, Lot N° XC345381, Invitrogen) overnight at 4°C, 1:200 dilution in an incubation solution. Samples were washed 3 times in PBS before being incubated with custom conjugated oligonucleotides as described below to AffiniPure goat anti-mouse IgG (H + L) secondary antibodies (Jackson ImmunoResearch, further details below) for 2 h at RT, 1:200 in incubation solution. Finally, samples were washed 3x in DNA-PAINT buffer.

### Custom Oligonucleotide-antibody conjugation

Docking strands were purchased from Integrated DNA Technologies (Coralville, Iowa, United States) with 5' azide and 3' Cyanine 3 dye modifications. Imagers were purchased from Eurofins Genomics (Ebersberg, Germany) with 3' ATTO 655 modification. All oligos were HPLC purified and shipped lyophilized. Oligos were reconstituted in PBS at 100 µM and stored at -20°C. The conjugation of oligo to antibody followed a click-chemistry protocol. Briefly, AffiniPure goat anti-Mouse IgG (H + L) secondary antibodies (115-005-003, Jackson ImmunoResearch) were incubated with DBCO-Sulfo-NHS ester (CLK-A124-10, Jena Bioscience) for 45 min. The reaction was stopped with 80 mM Tris-HCl for 10 min. The antibody-DBCO mixture was filtered using 7k MWCO Zeba Spin Desalting columns (Thermo Scientific) and the resulting antibody-DBCO was incubated with docking strand D2 (5' [AzideN] TTT TAG GTA AAT TTT GAT TGT GAG GAA G [Cy3], IDT), overnight at 4°C. Finally, the conjugation mixture was purified using Amicon Ultra-0.5 (UFC5100, Merck) and measured using an Implen-N60 touch spectrophotometer (Fisher Scientific) to confirm >1 oligo conjugation per antibody.

### Sample mounting for MINFLUX microscopy

For active sample-stabilization during MINFLUX experiments, 150 nm colloid gold beads (BBI Solutions, EM.GC150) were used, as fiducial markers[46]. Gold beads were added undiluted to the samples prior to imaging and allowed to settle for 5–10 min. Excess fiducials were removed with several washes in DNA-PAINT buffer. Once loose fiducials were removed, the chambers were filled with DNA-PAINT buffer containing nominally 1–2 nM Imager 3 ATTO 655 and made airtight with a glass coverslip placed over the chamber and sealed with Pinkysil (Barnes Products, Moorebank).

### MINFLUX 3D imaging

Samples were imaged on an Abberior Instruments MINFLUX microscope that was equipped with a 642 nm excitation laser. It could be run in confocal and MINFLUX imaging modes using a SLM-based beam shaping module and an EOD-based scanner for precision scanning

(used in MINFLUX mode), similar as previously described[47,48]. Emitted fluorescence signals were measured using two photon-counting avalanche photodiodes and suitable fluorescence filters (650–750 nm). In addition to MINFLUX modes, the microscope allows epifluorescence and 488 nm and 640 nm excitation confocal imaging acquisition. This was sometimes used to identify structures based on Alexa Fluor 647 nm labeling (when using sdABs conjugated to Alexa Fluor 647). To enable measurements with molecular precision, the Abberior MINFLUX microscope is equipped with a reflection-based stabilization unit, based on a 980 nm laser. It features real-time feedback that dynamically adjusts a 3-axis nano-positioning stage to fix the sample position with respect to the objective lens with standard deviations typically below 1 nm over minutes and up to hours under normal laboratory conditions[47] (if mechanical agitation of the microscope by external sources are kept at a reasonably low level). To compensate for changes in environmental conditions (especially temperature) that might manifest as a residual system drift, a beam line monitoring mode was activated (in addition to the simultaneously operating real-time feedback stabilization) by selecting several in-focus gold nanoparticles whose position is tracked over time. The particle-averaged tracking information from these nanoparticles is used to measure and correct drift by subtracting the apparent nanoparticle trajectory from the localization coordinates. Beam line monitoring can be maintained over hours and keeps drift induced position errors typically below 1.5 nm. The current implementation of beam line monitoring requires nanoparticles to be in focus so that the MINFLUX imaging volume in practice needs to be close to the coverslip interface during operation as most gold nanoparticles attach on the coverslip. The GFP signal from cultured cells were used to select suitable MINFLUX scan regions. For experiments involving myocytes the PMT signal was used to select cells that had adhered strongly to the coverslip, regions with good attachment appeared darker. Fast confocal scans, of potential regions of interest, were then taken using the 642 nm laser to excite the ATTO 655 imager. Binding events appeared stochastically as bright, long lasting, diffraction limited flashes confirming that the sample had been appropriately labeled with DNA-PAINT markers before acquiring MINFLUX data for several hours. The default 3D Imaging sequence provided by Abberior was used to determine the localizations of the fluorescent molecules.

## DNA-PAINT widefield imaging

Data was acquired using a modified Nikon Eclipse Ti-E inverted microscope (Nikon, Tokyo) with a fixed 60 × 1.49 NA APO oil immersion TIRF objective (Nikon, Tokyo)[49]. Images were captured with an Andor Zyla 4.2 sCMOS camera (Andor, Belfast) generally utilizing a 100 ms integration time, resorting to 25–50 ms when using fast DNA-PAINT imagers (with HEK293 RyR2$_{D4365-GFP}$ cells). Focus was controlled using a piezo objective scanner (P-725, Physik Instrument, Karlsruhe). For widefield fluorescence, a tuneable LED-light source (CoolLED, Andover) was used to aid cell selection and confirm proximity of 200 nm red fluorescent beads (fiducials) within the imaging region. These fluorescent markers were later used to correct for any lateral drift experienced over the course of data acquisition. Active z-stabilization was maintained whilst imaging through the use of an auxiliary camera monitoring transmitted light at a non-interfering wavelength from the super-resolution channel[2,49]. For comparison widefield DNA-PAINT imaging of HEK293 cells stably expressing RyR2-GFP, the 3D biplane configuration of the microscope was used. A continuous wave 647 nm diode laser (Omikron LuxX, Rodgau) was used to excite ATTO 655 imagers for single molecule measurements. Single molecule data was recorded using the open-source python software package called PyME[50] (Python Microscopy Environment, https://github.com/python-microscopy/python-microscopy). The imagers for widefield imaging of cardiomyocytes, either 'Imager 3' (Massive Photonics, undisclosed sequence) or A2 (5' TAC AAT CAA A

[ATTO 655], Eurofins), were diluted to 0.6 nM in DNA-PAINT buffer depending on whether the sdAb or the primary/secondary antibody system was used.

## 3D DNA-PAINT biplane widefield imaging

For 3D biplane imaging (used with HEK293 RyR2$_{D4365-GFP}$ cells and U2OS-Nup96-mEGFP cells) a splitter device positioned between the camera and microscope side port (Cairn, Australia), fitted with a 50/50 beam splitter dichroic and a weak lens ($f = 4$ m) placed in one of the paths, was employed. Prior to conducting biplane experiments a z-stack of ~200 images were taken of a field of view containing dark-red fluorescent 200 nm beads (ThermoFisher FluoSpheres F8807) using a step size of 50 nm. Beads were excited using an LED light source (p4000 CoolLED, Andover) emitting light nominally at 635 nm towards a Cy5 dichroic (Semrock, FF650) and the resulting fluorescence propagated through an appropriate emission filter (Semrock, LP02-647RU) towards the splitter device and camera. The z-stack was then used to obtain the point spread function (PSF) of the system through the opensource super-resolution microscopy and analysis software PyME and saved as a.tif file. Additionally, a vector map of the shifted position, between bead locations as viewed through channel 1 compared to channel 2, was acquired to digitally align the two channels in x and y with sub-pixel precision in PyME. This required a dense field of 200 nm dark-red fluorescent beads to be recorded, at a focal position approximately halfway between the two channels, as the sample was gradually moved in x and y directions until a completed map of the imaging field of view was constructed. Following these calibration steps biplane acquisition with real-time localization analysis was carried out under control of PyME.

## Analysis of 3D MINFLUX data

Raw MINFLUX localizations were exported from Abberior Instruments Imspector software[47] in.npy format and imported into PYMEVisualize[50] for further processing using custom developed open source plugins for MINFLUX data loading and processing. Raw MINFLUX localizations were processed by combining localizations to represent different aspects of the underlying structures, including docking strand locations and subunit locations. Docking strand locations were obtained from raw MINFLUX localizations to obtain locations that each represent a docking strand location during repeated localizations that are part of a single MINFLUX trace. To obtain unique subunit locations we used an approach similar to the procedure used previously to reassign filtered MINFLUX localizations to individual molecules[12]. These subunit locations were obtained by combining docking strand locations that arose from marker molecules that reappear due to repeated imager binding. Subunit locations were subjected to cluster analysis by DBSCAN clustering with the aim to characterize the underlying RyR2 clustering properties. Further details on the procedures to obtain the locations and clusters mentioned above are described in the supplementary methods. Data was visualized in two ways: (1) 3D views were obtained by OpenGL based rendering of MINFLUX localizations with coloring typically applied for a selected event property such as z-elevation, cluster size, cluster ID using the Pointsprite rendering mode of PYMEvisualize[50] which results in a Gaussian rendering in 3D. Alternatively, (2) images were rendered using the Gaussian Rendering function of PYMEvisualize[50] generally with a Gaussian sigma of 4 nm.

## Estimates of effective labeling efficiency from MINFLUX 3D NPC data

To estimate effective labeling efficiency we built on the approach established by Thevathasan et al.[19]. As in their method, we first detected NPCs in the super-resolution data and then went on to fit and align a template to the data, performed rotational alignment, determined how many segments were labeled (given the 8-fold symmetry of NPCs) and the resulting histogram was finally fit to a probabilistic

model. Given the high 3D resolution in MINFLUX 3D imaging with approximately isotropic resolution we extended the analysis by additionally implementing maximum-likelihood fitting of a 3D model[31] to precisely determine the 3D orientation of each NPC and separately evaluated labeling of nucleoplasmic and cytoplasmic segments of NPCs. Details of the full procedure are described in the supplementary methods and illustrated in Supplementary Fig. 4. Briefly, MINFLUX 3D data of U2OS-Nup96-mEGFP labeled with anti-GFP sdABs modified with DNA-PAINT docking strands was rendered to a 2D image and locations of NPCs were detected by a semi-manual approach in Fiji[51] and stored as regions-of-interest (ROIs). ROIs were read into the PYMEvisualize[50] SMLM data visualizer to generate a mask and label localizations belonging to each NPC with a unique ID. For each NPC the associated set of localizations were fit to a 3D double ring template using an algorithm similar to a previous approach[31] as detailed in the supplementary methods. The best-fit with a global maximizer yielded estimates of NPC geometry (diameter and vertical spacing) as well as the center and orientation of each NPC with high precision. This revealed distinct changes in orientation of NPCs across the field of view as illustrated in Supplementary Fig. 4C & D. Following rotational alignment, the number of labeled segments on the cytoplasmic and nucleoplasmic sides were determined for each NPC. The data from all NPCs in the dataset evaluated in this way were pooled into a cumulative histogram and fitted to a probabilistic model that predicts the relative number of 0 to 16 labeled segments as a function of a fit parameter $p_{LE}$ denoting the labeling efficiency. The complete fitting procedure was implemented in the form of plugins for PYMEvisualize. The best fit parameter $p_{LE}$ together with its uncertainty $\Delta p_{LE}$ are then provided as an estimate of effective labeling efficiency. Site visit counts were determined from the average number of localizations per segment and dividing this number by two as there are two target sites per segment. Comparisons using widefield SMLM biplane data of U2OS-Nup96-mEGFP labeled with anti-GFP sdABs modified with DNA-PAINT docking strands were analyzed in the same way.

### Blob analysis of widefield DNA-PAINT and MINFLUX DNA-PAINT data

Single molecule localization DNA-PAINT data were rendered using PYMEVisualize[50] into TIFF images with a pixel size of 5 nm by jittered triangulation[52]. A PyME object finding module (PYME.localization.ofind3d) was used to detect puncta within these images using a threshold of 1.0 and blur size 2.0. The 2D coordinates of the detected puncta were used to determine their nearest neighbor (NN) distances. Similarly, MINFLUX 3D datasets were rendered using PYMEVisualize[50] into TIFF images with a pixel size of 3 nm by Gaussian rendering. Rendering was performed from docking strand locations to benefit from the increased spatial resolution in MINFLUX data. Again, the PyME object finding module (PYME.localization.ofind3d) was used to detect puncta within these images using a threshold of 1.0 and blur size 2.0. The 2D coordinates of the detected puncta were used to determine their nearest neighbor (NN) distances and these were displayed in histogram form.

### Data display and statistical comparisons

Box plots show the median as center line, box limits are upper and lower quartiles, whiskers show 1.5x interquartile range and all data points are shown. In addition, the mean is shown as an open circle.

Statistics are reported as mean ± standard error of the mean unless noted otherwise. For individual fits of labeling efficiencies we report best fit value ± fit error.

For comparing nearest neighbor puncta distances, we conducted statistical comparisons implemented in the Python programming language within a Jupyter notebook environment. First, a one-way ANOVA test was performed using the scipy.stats module to assess the null hypothesis that the means of the three groups tested were equal.

The result suggested that at least one of the groups exhibited a significantly different mean compared to the others. Pairwise differences were further investigated using the statsmodels.api library utilizing the pairwise_tukeyhsd function. This function implements the Tukey's Honestly Significant Difference (Tukey HSD) test and compares all possible pairs of group means to identify statistically significant differences. The $p$-values were then recorded.

### Reporting summary

Further information on research design is available in the Nature Portfolio Reporting Summary linked to this article.

## Data availability

All source data and analysis scripts have been made available through figshare at the https://doi.org/10.6084/m9.figshare.23733579[53]. Raw data is available in the related figshare repository at https://doi.org/10.6084/m9.figshare.29196023[54]. The source data underlying all graphs shown are provided as a Source Data file. Source data are provided with this paper.

## Code availability

Processing was conducted using the PYME software package (https://github.com/python-microscopy/python-microscopy) with apps including PYMEVisualize and PYMEImage as well as by directly using PYME API functions in Jupyter notebooks (using Python 3) and the PYME plugin package PYME-extra (https://github.com/csoeller/PYME-extra). Associated custom analysis scripts are available at the https://doi.org/10.6084/m9.figshare.23733579[53]. Versions of python packages PYME-test-env, python-microscopy and PYME-extra used in this study are available via Zenodo DOIs https://doi.org/10.5281/zenodo.17768735[55], https://doi.org/10.5281/zenodo.17768283[56] and https://doi.org/10.5281/zenodo.17763621[57], respectively.

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

## Acknowledgements

We acknowledge project support from the Swiss National Science Foundation (SNSF 310030_208109 to C.S.). The equipment was supported by the SNSF (SNSF R'Equip 316030_213543 to C.S.), the University of Bern Innovation Fund and the Faculty of Medicine. P.P.J. acknowledges partial project support by the Health Research Council of New Zealand (HRC #20/370). W.E.L. was supported by the Norwegian Research Council (grant number 287395). We thank Jia Li for assistance with cell isolations in the preliminary stages of the study.

## Author contributions

Conceptualization: C.S, W.E.L, P.P.J. Data curation: C.S, I.J, A.H.C, A.F.E.B. Formal analysis: C.S, A.H.C. Funding acquisition: C.S. Investigation: A.H.C, R.J, I.J, A.M, E.L. Methodology: C.S, I.J, A.H.C, R.J, A.F.E.B, G.B. Project administration: C.S. Resources: R.J, G.B, A.F.E.B, I.J, P.P.J, W.E.L, C.S. Software: C.S, A.H.C. Supervision: C.S. Validation: R.J, A.M, A.H.C. Visualization: C.S, A.H.C. Writing: All authors contributed to the writing of the manuscript.

## Competing interests

I.J. declares the following competing interests. I.J. works at Abberior Instruments which develop and manufacture super-resolution fluorescence microscopes including the 3D-MINFLUX system used here. Remaining authors declare no competing interests.
