## [Transparent Peer Review file · Nature Communications]

MINFLUX microscopy resolves subunits of the cardiac ryanodine receptor and its 3D orientation in cells

Corresponding Author: Professor Christian Soeller

Version 0:

Reviewer comments:

Reviewer #1

(Remarks to the Author)

The authors used 3D MINFLUX to understand the distribution and orientation of RyR2 in cardiac myocytes. They employed single-domain antibodies (sdABs) to label RyR2 subunits with a precision of 3 nm in xyz.

Here are my major concerns:

1. In the abstract, via this statement,

"In practice, this capability is currently limited by a modest effective labeling efficiency (~10% subunit detection efficiency resulting in ~35% RyR2 labeling efficiency), which we measure in-situ using a novel procedure enabled by the true molecular resolution of MINFLUX microscopy."

The authors imply that the true molecular resolution of MINFLUX in some way helps them address the problem of labeling efficiency.

In light of the statement below from Gwosch et al. 2020 (which the authors cite), "With lens-based fluorescence microscopy having finally reached true nanometer resolution, it is important to bear in mind that fluorescence microscopes map nothing but the fluorophores; once the microscopes have fulfilled this task, they have accomplished their mission."

I would like to remind the authors that true molecular resolution doesn't solve the problem of labeling efficiency, and they should revise the sentence not to imply this implicitly.

2. Given the expertise of the authors in SMLM methods, and to really address the molecular resolution claim of MINFLUX, the author should do a comparison with any standard SMLM method and prove that indeed MINFLUX provides superior data.

3. A couple of recent papers have raised concerns regarding the consistency of MINFLUX data:

Prakash and Curd (2023) show that 2D and 3D MINFLUX data provide inconsistent quantitative data when the imaging modalities change.

<https://www.nature.com/articles/s41592-022-01694-x>

Prakash (2022) shows that MINFLUX data is less consistent and undersampled when compared to other standard SMLM datasets on nuclear pores samples

<https://royalsocietypublishing.org/doi/full/10.1098/rsta.2020.0145>

In light of both papers, a comparison with other SMLM methods is highly requested to gain confidence in the claims made in this paper.

4. In Figure 1, how do the authors establish the ground truth? Providing SMLM datasets with similarly tight distributions as in Figure 4 should be very convincing, especially when MINFLUX suffers from undersampling.

5. The FOV in Figure 1 is rather small, and I would like to see the full HEK293 cell. I realize MINFLUX doesn't provide large FOVs, so SMLM imaging would indeed be helpful here.

Overall, I request further SMLM datasets or other methods with similar resolution (like SIMFLUX) before I gain confidence in the MINFLUX data presented in the paper, which looks highly undersampled.

Reviewer #2

(Remarks to the Author)

MINFLUX microscopy resolves subunits of the cardiac ryanodine receptor and its 3D orientation in cells

Review Sept 23, 2023

In this work, the authors employed 3D MINFLUX microscopy to map the subunit structure of RyR2 homotetramers and their 3D orientation. The work exploited the use of single domain antibodies against FP-labelled domains of the RyR2 tetramer to provide 3D super-resolved localization data related to the RyR2 clustering, orientation, and structure. Coupling this with insights from complementary simulations, EM reference data, and DNA-PAINT experiments, the authors present a compelling workflow and approach that illustrates using this well-explored system how 3D MINFLUX could be applied to other similar molecular complexes and architectures. The work is well conducted and there are extensive reference experiments, simulations, and complementary data sets that are compelling as to the validity of this approach. The methods and materials are nicely described and the approaches appear to be quite robust and should be readily accessible to the community. There are a few questions / comments / updates that would make help strengthen this nice contribution

1. There are some nomenclature inconsistencies...

a. P. 3 – in the figure capture, the DBSCAN clustering uses “ $\epsilon=7$ nm” but later in the text on p4, “ ϵ ” is referred by its Greek letter

b. On at least one occasion DBSCAN isn't capitalized in the text – p. 7

2. On p. 8, the authors refer to FRC resolution – but the term : Fourier ring correlation – isn't defined in the text. It is however defined in the Supplementary materials

3. The formatting of the literature references is not consistent (some are in title case, others are in sentence case)

4. There is quite a bit of good information in the Supplementary Materials and it would be perhaps useful to see if more of the approaches could be include in the main body of the work. I was particularly interested in the details of the simulation work because it is key to interpreting and correlating with the experimental data and it would have been good if some details were included in the main text (or at least more explicit references to the Supplementary Materials section)

5. The authors provided a very important insight / perspective on P.10 when they emphasized the fact that the localization data is really for the label location on the protein, with an accounting for potential linkage effects. Moreover much of the interpretation relies heavily on a priori insights into the structure of the complex of interest, which in this case was provided by EM data. It would be interesting to expand this to a system that is perhaps less well organized or that exhibit a more complex and heterogeneous organization (i.e. mixture of known monomeric/dimeric/oligomeric species). It would be interesting to postulate some of the challenges that lie in those studies.

6. On P3 of the supplementary materials, the authors refer to “90 rotation” – I believe this should read “90 degree rotation”

Version 1:

Reviewer comments:

Reviewer #1

(Remarks to the Author)

I thank the authors for their response to my previous comments. The manuscript has undergone substantial revision, including the installation of a local MINFLUX microscope. The authors have also employed DNA-PAINT to improve detection probability.

As the authors rightly point out, it is important to distinguish between labeling efficiency and detection probability. My understanding is that DNA-PAINT helps with more detections but does not necessarily improve labeling efficiency unless a RESI type approach is used. If authors agree, this distinction should be made clearer in the text, else I am happy to hear their explanation.

A key aspect of scientific reporting is not only to highlight what works but also to communicate what does not. Much of the current work in the SMLM field remains two-dimensional. Therefore, it would be valuable for the community if the authors could explicitly state that in 2D, the MINFLUX data is broadly comparable to conventional SMLM. Unless I missed it in the main manuscript, I suggest adding this sentence either in the abstract or results section which authors mention in their cover letter.

“In x-y, the data appears broadly similar between both modalities once the MINFLUX data is laterally slightly blurred.”

Coming to 3D MINFLUX, in the cover letter, the authors state:

"As we have previously emphasized, the true improvement we observe with MINFLUX imaging is in the axial resolution, as clearly illustrated in SI Fig. 3D & E."

If this is indeed the case, I agree that it represents an advancement as previous studies have reported inconsistencies in 3D MINFLUX estimates, particularly when comparing 1-color vs. 2-color imaging modes (<https://www.nature.com/articles/s41592-022-01694-x>).

Given that the revised manuscript now focuses on 3D MINFLUX, I encourage the authors to present quantitative estimates of the axial separation between the nucleoplasmic and cytoplasmic rings of the nuclear pore complexes. For this, the authors may consider using publicly available tools such as:

1. The software from Jonas Ries (<https://www.nature.com/articles/s41592-022-01676-z>)
2. Alistair Curd's PERPL (<https://pubs.acs.org/doi/full/10.1021/acs.nanolett.0c03332>)
3. Other tools from Bernd Rieger/Sjoerd Stallinga, or John Daniai (ASAP, Nature Methods)

These tools are user-friendly and allow for modeling of single or multiple pores within a field-of-view, providing quantitative assessments. If the above is not feasible, the authors could select 15–20 well-resolved pores, perform line fitting, and report the axial spacing statistics.

I strongly recommend including a comparison with 3D SMLM datasets to contextualize the improvement provided by MINFLUX.

A discussion referencing von Appen paper or recent ones from Jan Ellenberg (<https://www.nature.com/articles/nature15381>) would also strengthen the discussion on nuclear pore structure.

In summary, I am not requesting additional lab experiments, but a quantitative comparison of 3D SMLM and MINFLUX that would serve as a minimal yet necessary benchmark to demonstrate that 3D MINFLUX results are indeed better and also consistent with current standards in the field.

Reviewer #2

(Remarks to the Author)

This is a substantially revised resubmission of the original manuscript. The authors have done an excellent job at addressing the key concerns of both reviewers and the work has been significantly improved. The authors' careful reconsideration and re-examination was well conducted and indeed presents a cautionary lesson for those applying these and related techniques..

After this adjustment, the work is well conducted and will be well received by the community.

Version 2:

Reviewer comments:

Reviewer #1

(Remarks to the Author)

I would like to commend the authors for their revisions and for the successful implementation of a commercial MINFLUX microscope along with the associated data analysis pipeline.

The manuscript now presents a coherent dataset. However, I would appreciate further clarification regarding the following:

1. While I appreciate the consistency shown between SMLM and MINFLUX results, the manuscript does not fully resolve the long-standing discrepancy between ~60 nm interlayer spacing reported in previous EM and SMLM studies versus the 50 nm value supported here. Notably, recent work by the Susan Cox lab (Rosten et al., 2025: <https://www.biorxiv.org/content/10.1101/2025.08.06.668903v1.full.pdf>) confirms the 60 nm interlayer distance between the cytoplasmic and nucleoplasmic rings, consistent with von Appen et al. and at odds with the findings reported in the current study.

Could the authors clarify from where this discrepancy arises? An explicit discussion would be valuable.

2. The following paragraph from the manuscript lacks specific information and could be interpreted as generic content: "Imaging of specific proteins that are part of the NPC has become both an important assay to quantify labeling efficiency..." I recommend rewriting this paragraph with concrete quantitative results. It would also be appropriate to cite Gwosch et al. (2020) here and their originally reported NPC measurements.

3. Given that many of the clustering results rely on a 100 nm cutoff and that comparable SMLM data already report similar patterns, it remains unclear why MINFLUX is necessary? Could the authors clarify what unique insights MINFLUX enables here that SMLM cannot, beyond improvements in some precision numbers?

4. If the methodological advances are primarily incremental, I encourage the authors to articulate more clearly what new biological insights emerge from this work. For instance, while the subunit orientation of RyR2 tetramers is described, the

functional implications of these structural findings are not evident. As someone less familiar with this specific biological system, the novelty and broader biological significance are not immediately clear from the current presentation.

5. I also think abstract, introduction, discussion can be further synced and polished. Presently, it is very difficult to read.

I thank the authors for their efforts and look forward to their responses.

Author response

We thank the reviewers for their insightful comments and helpful suggestions for improving our manuscript. The comments on detection efficiency prompted us to adapt our methodology to utilise DNA-PAINT instead of the originally used photoswitching (dSTORM) approach with MINFLUX. In doing so, it became clear that our previous data was indeed undersampled. We developed an assay to confirm the detection efficiency in situ and at the same time ensure that increased densities of localisations do reflect actual extra detected target sites (or more precisely, extra detected markers on target sites) rather than “repeat site visits” that could in principle result from using DNA-PAINT.

We thus repeated all our experiments in HEK293 cells and isolated myocytes with MINFLUX DNA-PAINT and present this new data for review. The main figures have all been updated to reflect this new, much higher quality data and we believe this addresses previous concerns. The difference in appearance is striking and quite apparent when inspecting the new data sets and in our view fully justified the redoing of all experiments. We thank the reviewers for bringing up the point of possible undersampling as the new data reveals the intricacies of RyR2 distribution that previously could not be resolved (which results from both spatial resolution and proper sampling).

Crucial for being able to redo these experiments was the delivery and installation of a MINFLUX microscope at our local facility and the ability to spend extensive time and effort on these experiments locally rather than travelling to a remote demo site as had been done for the initial data. As a consequence, completing this revised study has taken considerable time which was extended by some early hardware failures that we needed to rectify in close collaboration with the manufacturer.

To fully optimise the conditions we have introduced a refined and robust labelling efficiency assay and with that conducted to the best of our knowledge the first MINFLUX study with a focus on explicitly maximising the effective labeling efficiency, particularly the component that we term the “photo-detection probability” or “marker detection probability”. This reflects our observation that with dSTORM photoswitching and MINFLUX, at least in our hands, the dyes bleached very quickly (relative to the long scan times) while MINFLUX was searching for blink events to image so that many of the markers were never “sampled” by MINFLUX before the localisation rate declined. This could be avoided by judicious use of DNA-PAINT with MINFLUX where the assay is used to ensure that DNA-PAINT imaging has progressed to the point that all marker positions have been “harvested”. Site-loss in this case also helped ensure that markers were visited by successful imager localisations on average only once thus avoiding

overcounting. Sub-maximal photo-detection probability may indeed have hampered some previous MINFLUX studies.

REVIEWER COMMENTS

Reviewer #1 (Remarks to the Author):

The authors used 3D MINFLUX to understand the distribution and orientation of RyR2 in cardiac myocytes. They employed single-domain antibodies (sdABs) to label RyR2 subunits with a precision of 3 nm in xyz.

Here are my major concerns:

1. In the abstract, via this statement,

"In practice, this capability is currently limited by a modest effective labeling efficiency (~10% subunit detection efficiency resulting in ~35% RyR2 labeling efficiency), which we measure in-situ using a novel procedure enabled by the true molecular resolution of MINFLUX microscopy."

The authors imply that the true molecular resolution of MINFLUX in some way helps them address the problem of labeling efficiency.

In light of the statement below from Gwosch et al. 2020 (which the authors cite), "With lens-based fluorescence microscopy having finally reached true nanometer resolution, it is important to bear in mind that fluorescence microscopes map nothing but the fluorophores; once the microscopes have fulfilled this task, they have accomplished their mission."

I would like to remind the authors that true molecular resolution doesn't solve the problem of labeling efficiency, and they should revise the sentence not to imply this implicitly.

We thank the reviewer for their comment and are in full agreement with Gwosch et al that we are localising the position of our fluorophore. Thank you for highlighting that our abstract was perhaps not as clear as we had initially intended. We were trying to state that we had been able to estimate our labelling efficiency **because** of the improved isotropic resolution offered with MINFLUX microscopy. We were not intending to imply MINFLUX somehow improved labelling efficiency. We have adjusted our abstract to clarify this position, and it now reads:

"Combining MINFLUX with DNA-PAINT, to maximize detection efficiency, we measured in situ labeling efficiencies using NPC structures as reference and regularly achieved efficiencies around 50%, which would translate to RyR2 detection efficiencies close to 95%, i.e. the

probability that at least one subunit is detected, if target accessibility is similar. Using this approach, we detect dense and extended RyR2 expression in HEK293 cells with some clusters spanning several micrometres in extent and containing several hundred RyR2s.”

2. Given the expertise of the authors in SMLM methods, and to really address the molecular resolution claim of MINFLUX, the author should do a comparison with any standard SMLM method and prove that indeed MINFLUX provides superior data.

We thank the reviewer for the suggestion to include more SMLM data. We have taken this suggestion and included 2 new Supplementary Figures (SI2 and SI3) where we have imaged whole HEK293 cell overviews expressing RyR2s in 3D “widefield” DNA-PAINT using biplane. In x-y, the data appears broadly similar between both modalities once the MINFLUX data is laterally slightly blurred. As we have previously emphasized the true improvement that we have seen with MINFLUX imaging is the axial resolution and this is clearly illustrated in SI Fig 3D & E.

3. A couple of recent papers have raised concerns regarding the consistency of MINFLUX data:

Prakash and Curd (2023) show that 2D and 3D MINFLUX data provide inconsistent quantitative data when the imaging modalities change.

<https://www.nature.com/articles/s41592-022-01694-x>

Prakash (2022) shows that MINFLUX data is less consistent and undersampled when compared to other standard SMLM datasets on nuclear pores samples

<https://royalsocietypublishing.org/doi/full/10.1098/rsta.2020.0145>

In light of both papers, a comparison with other SMLM methods is highly requested to gain confidence in the claims made in this paper.

We agree that MINFLUX is a complex system and share a concern that users need to convince themselves as well as others that they are getting equivalent effective labelling efficiencies to simpler super-resolution approaches. In addressing our revisions, we have repeated our experiments using DNA-PAINT instead of the STORM based photo-switching data previously submitted. In addition, we have implemented a simultaneous NPC based Nup96 labelling assay to support that besides spatial resolution of marker distribution labelling efficiency similar to widefield SMLM is achievable and indeed achieved with the data presented here.

“In preliminary experiments with MINFLUX dSTORM photoswitching we had noticed relatively sparse apparent labeling which was greatly improved by use of MINFLUX

DNA-PAINT. We implemented a quantitative in situ effective labeling efficiency assay to confirm our qualitative observations as in MINFLUX DNA-PAINT care needs to be taken to fully collect localizations from the available marker distribution¹⁷....”

Whilst we observed both a qualitative and quantitative improvement in the density of labelling detection when using DNA-PAINT we wanted to go further and included a directly comparable in-situ assay to estimate effective labelling efficiencies. The nuclear pore complex and its regular symmetrical arrangement of the protein Nup96 provided the ideal sample to do this.

“We therefore co-cultured U2OS-Nup96-mEGFP cells with HEK293 cells stably expressing RyR2_{D4365-GFP} and then labeled these co-cultures with anti-GFP DNA-PAINT sdABs. We proceeded to record MINFLUX 3D images of Nup96-mEGFP labeling in U2OS-Nup96-mEGFP cells alongside labeling of RyR2-GFP in adjacent regions of interests (ROIs) in nearby HEK293 cells so that MINFLUX scanning multiplexed between these two ROIs, acquiring a dataset containing U2OS-Nup96-mEGFP 3D distribution adjacent to data of 3D staining of RyR2_{D4365-GFP} (Fig. 2A, compare also Fig 1). To estimate effective GFP labeling efficiency from the NPC 3D data we fitted a double ring template into imaged NPC structures to account for variations in 3D orientations of NPCs within the nuclear envelope (Fig. 2B). We used the detected and 3D aligned NPC localizations to measure the number of labeled segments (up to 16 in total with 8 on the cytoplasmic side and another 8 on the nucleoplasmic side). The obtained histograms were compared to the expected distributions for different labeling efficiencies to obtain an estimate of the effective GFP labeling efficiency p_{LE} . Further details of the template fitting procedure and the associated analysis are illustrated in Supplementary Fig. 4. For the NPC data shown in Fig. 2B this approach yielded $p_{LE} = 60.1\% \pm 0.2\%$ (Fig. 2C). On average, in such combined RyR2_{D4365-GFP} and U2OS-Nup96-mEGFP data sets, the effective GFP labeling efficiency p_{LE} was $51.2\% \pm 2.3\%$ (N=10 MINFLUX data sets from n=3 technical replicates, Fig. 2D). This confirmed that imaging durations were sufficient to fully capture the information from anti-GFP sdABs in the sample thus maximizing the marker detection probability P_{detect} .¹⁷ This required extensive imaging durations (from 5 to 11 h) to ensure capturing all chemically labeled locations in the sample and helped ensure that information from labeled RyR2_{D4365-GFP} was also fully captured, as the marker detection probability P_{detect} should be about equal regardless of the specific target as it mainly depends on the DNA-PAINT attachment.”

We have included a new main figure (Fig 2) that showcases this approach that we include below for ease of reference.

Figure 2. Determination of effective labeling efficiency of anti-GFP sdABs in co-cultures of cells expressing RyR2_{D4365-GFP} or U2OS-Nup96-meGFP. **A.** MINFLUX 3D image of two adjacent ROIs (green boxes) that were imaged time multiplexed in one imaging run. The left ROI was selected in an U2OS-Nup96-meGFP cell while the right one was located in a HEK293 cell stably expressing RyR2_{D4365-GFP}. Color indicates z-elevation. **B.** 3D view of MINFLUX localizations in the U2OS-Nup96-meGFP containing ROI with overlaid templates that identify locations and 3D orientation of NPCs. **C.** Quantitative analysis of NPC data shown in B by overlaying the cumulative histograms of labeled NPC “segments” (0-16) with predicted curves for various values of p_{LE} and line of best-fit (dashed) at $p_{LE} = 60.1 \pm 0.2 \%$. **D.** Across experiments on average an effective labeling efficiency of $51.2\% \pm 2.3\%$ was obtained (N=10 MINFLUX data sets from n=3 technical replicates). The inset shows the average number of site visits per U2OS-Nup96-mEGFP site, which is very close to 1 (1.18 ± 0.09 , N=10 MINFLUX data sets from n=3 technical replicates). Scale Bars A: 2 μ m, B: 200 nm.

Overall, our revised experiments reported a labelling density that increased by about a factor of 5 compared with the previous dSTORM based MINFLUX data as we show in the estimate included below to specifically illustrate this point. Note that we take care to ensure that there is no effect of “over-counting” in DNA-PAINT that could otherwise distort such density measurements.

Review Fig.1. Approximately five-fold increase in effective labeling density, consistent with an increase from ~10% in the dSTORM data to ~50% with the DNA-PAINT data. Note that we carefully monitored the number of site-visits that stay close to 1 as monitored by the NPC assay and shown in Fig. 2 D (inset) that is included in the figure above this one. Broadly speaking, for DNA-PAINT the site-loss that occurs with MINFLUX imaging ensures that no frequent revisits occur.

Overall, we believe this provides strong evidence that we have increased labelling-efficiency so that in combination with the spatial resolution afforded it does provide molecular resolution and in combination with the small size of the marker system does provide a sufficiently adequate representation of the underlying target distribution to deserve this qualifier.

We have also taken up the suggestion of the reviewer to stress that we are not “looking” at the sample directly but via the markers and imaging system we see a kind of replica that ideally is “essentially” equivalent to the underlying sample. We have added this to the discussion, stressing that we have paid explicitly attention to the crucial points when acquiring the revised data presented here.

“It is important to keep in mind that in fluorescence microscopy we only “see” dyes on markers, or in the case of DNA-PAINT, where an imager strand attaches to a docking strand on a marker, rather than the actual location of the protein targets themselves. In the scenario that (1) a substantial fraction of the target sites are occupied by markers with high specificity, (2) most or all of these markers are detected and precisely localized by the imaging system and (3) the resolved spatial detail is comparable (or not much smaller) than the marker size we can treat the obtained images as close replicas of the actual target site distribution. With the 3D resolution of MINFLUX, the high

specificity and comparatively small size of anti-GFP sdABs, the large size of the proteins involved and the focus on maximizing detection efficiency as described here the image data should provide an accurate reflection of the underlying RyR2 distributions in 3D. “

4. In Figure 1, how do the authors establish the ground truth? Providing SMLM datasets with similarly tight distributions as in Figure 4 should be very convincing, especially when MINFLUX suffers from undersampling.

We believe we have addressed the undersampling previously present by switching our single molecule approach to DNA-PAINT. Figure 1 in the main text has been updated with this new data and shows far denser clustering. This organisation was very similar to the super-resolution data we collected using a conventional “widefield” DNA-PAINT approach and can be seen in the new Supplementary Figures 2 and 3.

5. The FOV in Figure 1 is rather small, and I would like to see the full HEK293 cell. I realize MINFLUX doesn't provide large FOVs, so SMLM imaging would indeed be helpful here.

Many thanks for this suggestions which we have followed. We have included some full FOV for SMLM imaging in Supplementary Figures 2 and 3 that we hope the reviewer find useful in visualizing a full HEK293 cell.

Overall, I request further SMLM datasets or other methods with similar resolution (like SIMFLUX) before I gain confidence in the MINFLUX data presented in the paper, which looks highly undersampled.

Thank you again for your comments as they encouraged us to repeat our experiments with the DNA-PAINT approach now presented. We identified the MINFLUX marker detection efficiency as the limiting factor that in our hands is relatively poor with dSTORM photoswitching but can be effectively overcome with careful DNA-PAINT acquisition, ensuring to capture signal from most or all markers present in the sample. In addition, we believe the extra assay with direct effective labelling efficiency measurements we have conducted using NPCs and now include in this study should give greater confidence in the new DNA-PAINT MINFLUX data accompanying this manuscript.

Reviewer #2 (Remarks to the Author):

MINFLUX microscopy resolves subunits of the cardiac ryanodine receptor and its 3D orientation in cells

Review Sept 23, 2023

In this work, the authors employed 3D MINFLUX microscopy to map the subunit structure of RyR2 homotetramers and their 3D orientation. The work exploited the use of single domain antibodies against FP-labelled domains of the RyR2 tetramer to provide 3D super-resolved localization data related to the RyR2 clustering, orientation, and structure. Coupling this with insights from complementary simulations, EM reference data, and DNA-PAINT experiments, the authors present a compelling workflow and approach that illustrates using this well-explored system how 3D MINFLUX could be applied to other similar molecular complexes and architectures. The work is well conducted and there are extensive reference experiments, simulations, and complementary data sets that are compelling as to the validity of this approach. The methods and materials are nicely described and the approaches appear to be quite robust and should be readily accessible to the community. There are a few questions / comments / updates that would make help strengthen this nice contribution

1. There are some nomenclature inconsistencies...

a. P. 3 – in the figure capture, the DBSCAN clustering uses “ $\epsilon=7$ nm” but later in the text on p4, “ ϵ ” is referred by its Greek letter

b. On at least one occasion DBSCAN isn't capitalized in the text – p. 7

We thank the reviewer for spotting these inconsistencies in our text. We have carefully gone through our revised manuscript to correct these errors. We now use ϵ and capitalised DBSCAN throughout.

2. On p. 8, the authors refer to FRC resolution – but the term : Fourier ring correlation – isn't defined in the text. It is however defined in the Supplementary materials

We have made sure to define the acronyms FRC and FSC in the main text.

3. The formatting of the literature references is not consistent (some are in title case, others are in sentence case)

Thank you for making us review our referencing, we spotted a couple of additional inconsistencies in our formatting which we have now corrected.

For the specific upper and lower case inconsistencies, we checked the following two references that are “title case”:

- Cabra, V., Murayama, T. & Samsó, M. Ultrastructural Analysis of Self-Associated RyR2s. *Biophys. J.* 110, 2651–2662 (2016).
- Hiess, F. et al. Distribution and Function of Cardiac Ryanodine Receptor Clusters in Live Ventricular Myo-cytes. *J. Biol. Chem.* 290, 20477–20487 (2015).

These are capitalised but apparently directly reflect the cited journal policy and are capitalised on the journal websites. As such we believe it would be correct to 100% reproduce the published title rather than impose lowercase for consistency, but we defer to the editors as to their preference.

4. There is quite a bit of good information in the Supplementary Materials and it would be perhaps useful to see if more of the approaches could be include in the main body of the work. I was particularly interested in the details of the simulation work because it is key to interpreting and correlating with the experimental data and it would have been good if some details were included in the main text (or at least more explicit references to the Supplementary Materials section)

We thank the reviewer for their suggestion. From our own concerns about possible undersampling and those additionally raised by Reviewer 1 we decided to repeat our experiments using DNA-PAINT instead of the photoswitching STORM type experiments we previously submitted. With the improved detection efficiencies that we now achieve we obtain far denser clusters. In accomplishing this it made automating a detection procedure of tetramers, trimers etc currently very difficult for us to implement. We believe this would have to include the maximum-likelihood template fittings as we have implemented for NPC data. This is very CPU intensive and time consuming for large RyR datasets and we have not yet succeeded in generating a computationally feasible implementation. We have added the below text to the manuscript and hope ourselves or others can make advances in an automatic assignment so as to be able to calibrate effective labelling efficiencies from the target itself some time in the future.

“Due to the relatively dense labeling detected in the MINFLUX DNA-PAINT datasets with optimized DNA-PAINT acquisition (Fig. 2) automatic detection of candidates for such directly labeled adjacent subunits proved difficult. Often it was not possible to unequivocally assign subunit positions to individual candidate receptor positions in densely stained regions. In principle, such an automated assignment would enable estimating the effective labeling efficiency in situ from the fractions of receptors with 1 to maximally 4 labeled subunits and provide an alternative to the use of NPC based calibration.”

5. The authors provided a very important insight / perspective on P.10 when they emphasized the fact that the localization data is really for the label location on the protein, with an accounting for potential linkage effects. Moreover much of the interpretation relies heavily on a priori insights into the structure of the complex of interest, which in this case was provided by EM data. It would be interesting to expand this to a system that is perhaps less well organized or that exhibit a more complex and heterogeneous organization (i.e. mixture of known monomeric/dimeric/oligomeric species). It would be interesting to postulate some of the challenges that lie in those studies.

We thank the reviewer for their comment and agree this is an important perspective to keep in mind when doing any fluorescent imaging especially at the molecular level. With the denser clustering we observed when applying DNA-PAINT our previous labelling efficiency method proved more difficult to implement. As such, and additionally inspired by your comment, we opted to demonstrate that a co-culture approach could be used to calibrate and determine effective labelling efficiency of anti-GFP sdABs. We did this using NPCs which have a known arrangement of the Nup96 protein. This has led to a new Main Figure 2 and an additional Supplementary Figure 4 demonstrating the process which we include below for ease of reference. Critical for robustness is the template fitting approach we describe as part of the NPC calibration procedure. Even for a regular oligomer such as RyR2 applying the template fitting approach with dense labelling where it is not always clear how subunits group into candidate RyRs this becomes rapidly computationally impractical as template fitting involves “slow” global optimization.

At this stage the challenge to translate this to a less well organized and potentially structurally less well known scenario remains a challenge that may well be accessible using statistical methods and possibly machine learning but we felt it would be too speculative to include without direct data in this MS.

We have added a line to our discussion to emphasize this idea:

“In principle, the characteristic configuration of 4 subunits on RyR2s should also enable a direct way of estimating the effective labeling efficiency but it proved difficult to unequivocally assign subunit positions to candidate receptor positions in the densely stained samples. A template fitting approach similar to the NPC assay is desirable but complicated by identifying candidate subunit groupings to fit to. Fitting to all possible groupings is currently prohibitively CPU intensive due to the global minimization essential in template fitting. An alternative statistical approach or accelerating analysis with a suitable machine-learning based method may circumvent this and also eventually allow extension to more heterogenous multi-unit complexes.”

Supplementary Figure 4. Determination of effective labeling efficiency with NPC structures obtained from U2OS-Nup96-mEGFP labeling. **A.** A 2D rendering of the MINFLUX 3D data was rendered and processed in Fiji to detect NPC structures. **B.** Localizations belonging to identified NPC structures were fit using a maximum-likelihood approach implemented with a double ring template. The top row shows various views of the unaligned data from a single NPC prior to fitting and the bottom row indicates the best fit transformation aligning the data with the template. **C.** The best fit data was used to generate template overlays (double ring plus a line indicating the central symmetry axis through the pore). **D.** In this larger field-of-view, otherwise as in C, the variation in NPC axis alignment across the field is visible. **E.** Labeling of segments was determined to generate an experimental histogram (i) following rotational alignment of fitted data to segment boundaries (ii). The aligned NPC localizations can also be overlaid to produce and render an average distribution of Nup96 localizations (iii) which clearly exhibits the 8-fold symmetry and also shows ~12nm distal peaks resulting from the two Nup96 sites in a segment. **F.** Fitting of the cumulative experimental histogram (dots) to the model yields an estimate of effective labeling efficiency as best fit (dashed line), here 61.3% \pm 0.1%. **G** and **H.** Testing of labeling threshold against background contribution as described in text, supporting the suitability of a threshold of 1. Scale bars A, D, G: 200nm, C: 100 nm, E: 50 nm.

6. On P3 of the supplementary materials, the authors refer to “90 rotation” – I believe this should read “90 degree rotation”

Thank you, this section has now been removed to reflect the new data we present.

Point-by-point responses to the reviewers' comments

We thank both reviewers for their efforts in reading our revised manuscript and providing constructive comments for further improvement which we address below.

REVIEWER COMMENTS

Reviewer #1 (Remarks to the Author):

I thank the authors for their response to my previous comments. The manuscript has undergone substantial revision, including the installation of a local MINFLUX microscope. The authors have also employed DNA-PAINT to improve detection probability.

We thank the reviewer for their in-depth constructive assessment of our work and address their additional remarks below.

As the authors rightly point out, it is important to distinguish between labeling efficiency and detection probability. My understanding is that DNA-PAINT helps with more detections but does not necessarily improve labeling efficiency unless a RESI type approach is used. If authors agree, this distinction should be made clearer in the text, else I am happy to hear their explanation.

The reviewer is absolutely correct that we expect no marked difference in the 'chemical' labelling efficiency (P_{chem}) when using the same marker harbouring either a STORM dye or a DNA-PAINT docking sequence. Rather, it seems that the number of successful detections is increased with DNA-PAINT, as the reviewer suggests. We have added the following sentences to our manuscript to explicitly state this:

In the Results section (lines 93-95): "We attribute this not to a better chemical labeling efficiency but more likely, in our hands, to a higher effective photo-detection probability of chemically bound markers with DNA-PAINT".

And in the Discussion (lines 299-305): "Similar to previous reports, we observed an apparently quite small effective labeling efficiency when we utilized dSTORM based dye blinking in our preliminary MINFLUX experiments. We hypothesized that alongside photobleaching this resulted from stringent "photo-detection" requirements in MINFLUX imaging, in addition to the purely "chemical" labeling (P_{chem}) efficiency of labeling with the sdABs. The P_{chem} of a sdAb is not expected to significantly differ between markers harboring either a photoswitchable dye or a DNA-PAINT docking strand. However, the "photo-detection" requirement should be easier to fulfil with MINFLUX DNA-PAINT imaging since fresh imager molecules can diffuse in and bind to docking strands on sdABs."

A key aspect of scientific reporting is not only to highlight what works but also to communicate what does not. Much of the current work in the SMLM field remains two-dimensional. Therefore, it would be valuable for the community if the authors could explicitly state that in 2D, the MINFLUX data is broadly comparable to conventional SMLM. Unless I missed it in the main manuscript, I suggest adding this sentence either in the abstract or results section which authors mention in their cover letter.

“In x–y, the data appears broadly similar between both modalities once the MINFLUX data is laterally slightly blurred.”

We agree with the sentiments of the reviewer and have altered lines 83–86 of the main manuscript which now read: “When compared at the slightly lower lateral spatial resolution available in wide-field DNA-PAINT, lateral x-y views look broadly similar in both modalities once the MINFLUX data is laterally slightly blurred, albeit with a very notable improvement in z-resolution in the 3D MINFLUX data (Supplementary Figs. 2 and 3)”

And in the Discussion in lines 312–314 that read: “This is also consistent with the qualitatively similar appearance of MINFLUX and widefield DNA-PAINT RyR2 labeling when compared in 2D and at the lower resolution available with widefield super-resolution imaging (Supplementary Fig. 3).”

Coming to 3D MINFLUX, in the cover letter, the authors state:

“As we have previously emphasized, the true improvement we observe with MINFLUX imaging is in the axial resolution, as clearly illustrated in SI Fig. 3D & E.”

If this is indeed the case, I agree that it represents an advancement as previous studies have reported inconsistencies in 3D MINFLUX estimates, particularly when comparing 1-color vs. 2-color imaging modes (<https://www.nature.com/articles/s41592-022-01694-x>).

Given that the revised manuscript now focuses on 3D MINFLUX, I encourage the authors to present quantitative estimates of the axial separation between the nucleoplasmic and cytoplasmic rings of the nuclear pore complexes. For this, the authors may consider using publicly available tools such as:

1. The software from Jonas Ries (<https://www.nature.com/articles/s41592-022-01676-z>)
2. Alistair Curd’s PERPL (<https://pubs.acs.org/doi/full/10.1021/acs.nanolett.0c03332>)
3. Other tools from Bernd Rieger/Sjoerd Stallinga, or John Danial (ASAP, Nature Methods)

These tools are user-friendly and allow for modeling of single or multiple pores within a field-of-view, providing quantitative assessments. If the above is not feasible, the authors could select 15–20 well-resolved pores, perform line fitting, and report the axial spacing statistics.

I strongly recommend including a comparison with 3D SMLM datasets to contextualize the improvement provided by MINFLUX.

We thank the reviewer for this excellent suggestion and the opportunity to provide additional data to strengthen our manuscript. Indeed, we had already employed one of the approaches referred to by the reviewer (Jonas Ries <https://www.nature.com/articles/s41592-022-01676-z>) and implemented the algorithms described therein within the PYME environment so that we could carry out the quantitative analysis to estimate labeling efficiencies precisely (as shown in Fig. 2 and Supplementary Fig. 4 in the original revision). As a by-product we also obtain the geometric parameters emphasized by the reviewer, particularly the ring spacing. As suggested, we have now extended this analysis to SMLM data of NPCs which we had already obtained as part of our SMLM biplane experiments. This has culminated in the addition of a new supplementary figure, (Supp. Fig. 5) that we include below for ease of reference, where we compare 3D MINFLUX and 3D widefield SMLM NPC datasets including the extracted geometries obtained in both modalities. This comparison suggests an improved estimate is achievable with

the MINFLUX approaches that we describe in this MS, taking care to maximize labeling and its successful imaging.

The additional method description for the NPC geometry analysis has been added to the supplementary information document and starts there on page 5.

We have inserted a reference to this figure on lines 112-116 in the main text and describe the main conclusion there as follows:

“In addition, we analyzed the geometry of NPCs and obtained the expected ring spacing of ~50 nm (Supplementary Fig. 5). Consistent with the comparison between MINFLUX and SMLM biplane data of RyRs (Supplementary Fig. 3) MINFLUX based NPC analysis benefited from the high axial resolution in 3D MINFLUX imaging and provided more precise geometry parameter estimates as compared to widefield SMLM data (Supplementary Fig. 5 C-E).”

Supplementary Figure 5. Analysis of NPC geometries obtained with MINFLUX and SMLM 3D imaging. **A.** Gallery of Nup96-EGFP labeled NPCs from a MINFLUX DNA-PAINT dataset in lateral (x-y) and axial (x-z) views. **B.** Equivalent gallery from a SMLM biplane DNA-PAINT dataset in lateral (x-y) and axial (x-z) views. **C.** Distribution of “ring” diameter $r = 108.9 \pm 2.7$ nm (mean \pm standard deviation, 82 NPCs from one MINFLUX 3D dataset) and ring spacing $s = 47.2 \pm 2.7$ nm (mean \pm standard deviation, 82 NPCs from one MINFLUX 3D dataset) obtained from MINFLUX data. **D.** Distribution of “ring” diameter $r = 114.9 \pm 3.4$ nm (mean \pm standard deviation, 87 NPCs from one SMLM biplane 3D dataset) and ring spacing $s = 53.4 \pm 6.8$ nm (mean \pm standard deviation, 87 NPCs from one SMLM biplane 3D dataset) from SMLM data. **E.** Comparison of mean ring diameters and ring spacings measured with MINFLUX or SMLM images; MINFLUX derived mean ring diameter $r = 111.9 \pm 2.2$ nm (mean \pm standard deviation, 5 MINFLUX datasets from N=2 independent realizations, 365 NPCs in total) vs SMLM mean ring diameter $r = 117.9 \pm 4.1$ nm (mean \pm standard deviation, 4 SMLM datasets from N=2 independent realizations, 228 NPCs in total); MINFLUX derived mean ring spacing $s = 49.6 \pm 1.9$ nm (mean \pm standard deviation, 5 MINFLUX datasets from N=2 independent realizations, 365 NPCs in total) vs SMLM mean ring spacing $s = 58.9 \pm 3.9$ nm (mean \pm standard deviation, 4 SMLM datasets from N=2 independent realizations, 228 NPCs in total). **F.** Averaged 3D NPC Nup96 images from an analyzed MINFLUX dataset (top row, N = 29 NPCs, 1396 combined localizations) versus an averaged 3D NPC Nup96 image from a SMLM dataset (bottom row, N = 72 NPCs, 4512 combined localizations). Lateral views of the nucleoplasmic side (x-y) and axial side-views (x-z) are shown. ~ 12 nm distal peaks resulting from adjacent Nup96 sites are indicated in the MINFLUX nucleoplasmic view (arrows). The center line in the boxplots show the median and the large circular point is the mean of the datapoints displayed. Scale bars: 50 nm.

A discussion referencing von Appen paper or recent ones from Jan Ellenberg (<https://www.nature.com/articles/nature15381>) would also strengthen the discussion on nuclear pore structure.

We thank the reviewer for this suggestion and agree that discussing these papers and the influence of super-resolution microscopy on determining NPC structure and function in situ would strengthen the general discussion. We have therefore added a section to the discussion of the nuclear pore complex, lines 336-347, and included the suggested references:

“NPC imaging with MINFLUX microscopy

Imaging of specific proteins that are part of the NPC has become both an important assay to quantify labeling efficiency¹⁸ and has greatly improved understanding of the 3D architecture of the NPC and the role, placement and dynamic configuration of various proteins in the NPC. Important insights have been achieved by combining cryo-electron tomography with mass spectrometry, biochemical analysis, perturbation experiments and structural modelling to produce a comprehensive architectural model of the human nuclear pore complex³¹. Super-resolution imaging is increasingly being used to augment the information about structure and function of the nuclear pore complex^{18,30,32,33} with MINFLUX microscopy playing an increasingly important role^{10,14} due to the high spatio-temporal resolution that is accessible. This includes most recently the dynamic monitoring of nuclear import and export using MINFLUX tracking capabilities while also structurally monitoring the NPC scaffold in live cells³⁴. Here we exploited the stereotypical appearance of NPCs to quantify effective labeling efficiency and additionally used it to extract NPC geometry parameters with high precision.”

In summary, I am not requesting additional lab experiments, but a quantitative comparison of 3D SMLM and MINFLUX that would serve as a minimal yet necessary benchmark to demonstrate that 3D MINFLUX results are indeed better and also consistent with current standards in the field.

We thank the reviewer again for their suggestions to further improve and clarify specific elements of our manuscript. We hope that with these additions we have addressed the remaining concerns of the reviewer.

Reviewer #2 (Remarks to the Author):

This is a substantially revised resubmission of the original manuscript. The authors have done an excellent job at addressing the key concerns of both reviewers and the work has been significantly improved. The authors' careful reconsideration and re-examination was well conducted and indeed presents a cautionary lesson for those applying these and related techniques..

After this adjustment, the work is well conducted and will be well received by the community.

We thank the reviewer for their time and effort reviewing our MS and for providing informative and constructive feedback.

Point-by-point responses to the reviewers' comments

Reviewer Comments

Reviewer #1 (Remarks to the Author):

I would like to commend the authors for their revisions and for the successful implementation of a commercial MINFLUX microscope along with the associated data analysis pipeline.

I thank the authors for their efforts and look forward to their responses.

We thank the reviewer for their constructive assessment and address their additional remarks below.

The manuscript now presents a coherent dataset. However, I would appreciate further clarification regarding the following:

1. While I appreciate the consistency shown between SMLM and MINFLUX results, the manuscript does not fully resolve the long-standing discrepancy between ~60 nm interlayer spacing reported in previous EM and SMLM studies versus the 50 nm value supported here. Notably, recent work by the Susan Cox lab (Rosten et al., 2025: <https://www.biorxiv.org/content/10.1101/2025.08.06.668903v1.full.pdf>) confirms the 60 nm interlayer distance between the cytoplasmic and nucleoplasmic rings, consistent with von Appen et al. and at odds with the findings reported in the current study. Could the authors clarify from where this discrepancy arises? An explicit discussion would be valuable.

2. The following paragraph from the manuscript lacks specific information and could be interpreted as generic content:

"Imaging of specific proteins that are part of the NPC has become both an important assay to quantify labeling efficiency..."

I recommend rewriting this paragraph with concrete quantitative results. It would also be appropriate to cite Gwosch et al. (2020) here and their originally reported NPC measurements.

We respond to both points 1 and 2 here. Prompted by point 1 of the reviewer we have reviewed the literature which revealed that (1) the von Appen study structural model predicts a ring spacing of ~57 nm for the relevant Nup96 residues, (2) that a broad range of super-resolution studies has reported values close to 50 nm spacing and that (3) a recent cryoEM study of the NPC architecture in native tissue predicts a lower height of the NPC in the native tissue context than the von Appen study and puts this down to the fact that previous studies, including the von Appen study, at least partially rely on purified nuclear envelope preparations. Finally, (4) given the physical offset of the labeling system in our and a previous super-resolution study, the agreement between distances is very reasonable. Point (3) is broadly consistent with the slightly smaller value for the ring spacing from super-resolution studies (including 2 MINFLUX studies) as these are all imaged in the native cell/tissue context. The preprint cited by the reviewer above has reanalysed existing super-resolution datasets from prior publications which originally did not report the ring spacing. In this context we note that the measured z ring spacing sensitively

depends on the z-axis scaling correction which results from refractive index mismatch. This correction factor is difficult to determine accurately and differences in this calibration could explain some of the differences between studies. In addition, according to point (3) above, the NPC geometry is plastic in the experimental context and differences in sample preparation may contribute to subtle (~10%) differences.

As suggested by the reviewer in point 2, we have therefore rewritten the corresponding discussion paragraph and essentially present the detailed quantitative review as given above, with all relevant references.

Finally, we emphasize that for the purposes of our study presented here, the primary goal of the Nup96 signal analysis is a quantification of labeling efficiency. With the maximum-likelihood methods we employ and the high spatial resolution in the MINFLUX datasets, the labeling efficiency analysis is robust against small variations in NPC geometry.

For ease of reference, we reproduce the relevant section of the revised discussion section here:

“Imaging of the NPC has become an important assay to quantify labeling efficiency¹⁸ and has contributed to revealing the 3D architecture of the NPC in situ. By combining cryo-electron tomography with mass spectrometry, biochemical analysis, perturbation experiments and structural modelling a comprehensive architectural model of the human nuclear pore complex has been previously generated³¹. From this model, the spacing between cytosolic and nucleosolic rings of Nup96 protein locations was obtained as ~57 nm³². Super-resolution imaging has since then been used to estimate these dimensions from fluorescence data with widefield SMLM based estimates of Nup96 ring spacing of 49.8¹⁸, 50.2³⁰ and 51.2 nm³³. The first MINFLUX measurement of this distance determined 46 nm¹⁰ and a recent live cell MINFLUX study estimated a ring spacing of 51.5 nm³⁴. In our experiments we measured a ring spacing of 49.6 nm (Supplementary Fig. 5), similar to previous super-resolution based measurements. A recent preprint reanalyzed published datasets and determined ring spacings closer to the predicted distance from the von Appen cryo-EM study³¹, namely 57.5 nm, and a comparatively large value of 63.2 nm (from a 4Pi-STORM dataset)³⁵.

The approximately 10% difference between most reported super-resolution and the original cryoEM measures may be explained by (a) the finding that NPCs exhibit substantial plasticity and that the purification of the nuclear envelope as employed in prior cryoEM studies may influence the human NPC structure³⁶, (b) that the reported ring spacing in super-resolution studies sensitively depends on the z-axis correction that is applied to correct for refractive index mismatch¹⁰, and (c) that some offsets may result from the labeling system employed. In accordance with the frequently reported ring spacing of ~50 nm in super-resolution studies, recent cryo-EM data of NPCs in the native cellular context yielded a reduced NPC height (by ~10%) compared with previous models that used data from purified samples³⁶. Both in a previous MINFLUX study and our study an sdAB labeling system against GFP in Nup96-mEGFP was employed, with an up to 6–7 nm distance between the dye and the NUP96 attachment point³³. Given these sources of differences between studies all reported values agree comparatively well. For the purposes of this study the focus has been on the ability to quantify effective GFP labeling efficiencies. With the maximum-likelihood fitting approach employed here labeling efficiency estimates are robust against small (~10%) geometry variations.”

3. Given that many of the clustering results rely on a 100 nm cutoff and that comparable SMLM data already report similar patterns, it remains unclear why MINFLUX is necessary? Could the authors clarify what unique insights MINFLUX enables here that SMLM cannot, beyond improvements in some precision numbers?

As we describe in several places and have further emphasized in the revised version of the manuscript, while the 100 nm cutoff is a reasonable choice to identify clusters that act as calcium release units (as the RyR2s contained often open in unison upon cluster triggering), the

excitability and functional control of calcium release depends on the neighbor arrangement of RyR2s on a scale of ~10 nm, as well as their relative orientation to each other. This high detail information is only revealed with MINFLUX and qualitatively goes far beyond information available from previous widefield SMLM data.

In addition, the MINFLUX data has helped resolve a discrepancy between previous SMLM data and EM data. The MINFLUX 3D data revealed a higher density of RyR2s than previous SMLM data, consistent with the few available volume EM data (Fig.6 and Supplementary Fig. 9). As the reviewer states in their other comments, resolving potential inconsistencies between super-resolution and EM data is regarded as important by them in their field revolving around the NPC structure.

In the revised version, we have as requested further emphasized these points at the end of the introduction and also revised a section of the discussion to focus on these points explicitly and state them clearly.

Specifically, at the end of the introduction we now write:

“Our results demonstrate that MINFLUX 3D imaging of RyR2s combined with a small marker approach can determine RyR2 3D orientations in intact cells. We show, to the best of our knowledge, the first fully isotropic 3D light microscopy data of RyR2s in HEK293 cells and mouse ventricular myocytes where localization precision is better than 3 nm in all directions. We demonstrate how to optimize labeling and detection in MINFLUX 3D imaging and achieve 50% effective labeling efficiency for GFP tagged subunits, implying a homo-tetramer RyR2-GFP detection probability approaching 95%. The high density of RyR2 observed in large clusters in mouse myocytes resolves the apparent discrepancy between super-resolution and electron tomography data as we show that densities of RyR2s in large clusters are comparable between modalities if we assume a subunit detection efficiency of ~50%. Given increasing evidence that the neighbor arrangement of RyR2s on a scale of ~10 nm affects the coordination of RyR2 openings and cluster excitability¹³, MINFLUX 3D based imaging of RyR2s provides an important assay for mechanistic studies of pathophysiological changes in the control of calcium release in cardiac myocytes.”

And in the discussion:

“A key goal in grouping RyR2s into clusters is to identify groups of receptors that activate in unison to give rise to Ca²⁺ sparks and similar local release events. Due to calcium-induced calcium release (CICR) these functional groups are generally formed by RyR2s in close proximity and thus approaches to group RyR2s into clusters according to a distance criterion have been widely used. While the DBSCAN algorithm may not be optimal for detecting clustering in some biological scenarios³⁹, for CICR it accurately reflects the mechanism of fire-diffuse-fire activation of neighboring RyR2s and was therefore used with a 100 nm distance to identify groupings of receptors likely to activate in unison^{3,22,40}. Analysis for other signaling distances is possible by adjusting the clustering parameter ϵ . Indeed, with this approach we identified very extended RyR2 clusters in the HEK293 model employed here. Similarly, large RyR2 clusters dominate the visual appearance of RyR2 distribution in ventricular myocytes (containing typically many tens to >100 RyR2s). From a Ca²⁺ signaling point of view we would expect these clusters to dominate the release during activation and also generate the majority of the spontaneous Ca²⁺ sparks observed in otherwise quiescent myocytes, a major component of the functionally important SR leak in these cells⁴¹.”

A significant advance of MINFLUX 3D imaging compared to previous super resolution modalities is the ability to identify the relative placement and orientation of single RyR2 channels within a cluster. Whilst cluster size was previously used to define the activity of clusters it is clear from more recent EM data that things are more complex⁴, as (patho)physiological stimuli can alter the nanometer scale packing of channels within the cluster. Mathematical modeling has indicated that these changes critically alter both the fidelity and time course of spark generation^{7,5}. Compared to EM based 3D

methods, sample preparation for MINFLUX imaging is comparatively straightforward. It can routinely achieve single channel localization and orientation to enable probing of the plastic geometry of channels within a cluster, under a range of conditions, with good statistical coverage. Therefore, MINFLUX 3D imaging will be crucial to expand our knowledge of how intra-cluster geometry influences cluster function. Looking ahead, it is capable of multi-channel imaging and will also allow for simultaneous imaging of local cellular compartment architecture. In conjunction with such molecular resolution data and mathematical modelling as well as correlative functional studies^{9,8,42}, MINFLUX 3D imaging lays out a clear path to arrive at more precise and functionally relevant criteria to determine RyR2 functional groupings and refine our understanding of the RyR2 cluster concept for cardiac Ca²⁺ signaling.”

4. If the methodological advances are primarily incremental, I encourage the authors to articulate more clearly what new biological insights emerge from this work. For instance, while the subunit orientation of RyR2 tetramers is described, the functional implications of these structural findings are not evident. As someone less familiar with this specific biological system, the novelty and broader biological significance are not immediately clear from the current presentation.

As we point out in response to point 3 above, the methodological advancement in combining MINFLUX 3D, high effective labeling efficiency and using molecular (GFP/RFP) markers on RyR2 subunits has importance for the understanding of cardiac calcium regulation in the heart in (patho-)physiology. We specifically refer to the importance of the subunit orientation of RyR2 tetramers as a determinant of the coordination of RyR2 openings and cluster excitability. As mentioned in the response to point 3 we have made these points prominently in the revised abstract, introduction and discussion.

5. I also think abstract, introduction, discussion can be further synced and polished. Presently, it is very difficult to read.

We have revised and polished the abstract which we include below for reference. In addition, as pointed out in responses to points 1-3 we have improved and further polished the introduction and discussion aiming for clarity with a non-expert audience in mind.

We trust this now constitutes a cohesive manuscript with high relevance for a broad audience.

The revised abstract, for ease of reference is included here:

“The cardiac ryanodine receptor (RyR2) constitutes the molecular basis of the process of calcium-induced calcium release where activation of RyR2s can be locally regenerative. Here, we present purely optical data of RyR2 distribution with sub-molecular resolution by applying 3D MINFLUX microscopy. Using single-domain antibodies and DNA-PAINT we determine the location of individual RyR2 subunits with high precision (~3 nm) and resolve the 3D orientations of RyR2s in-situ. We measured labeling efficiencies of ~50%, implying RyR2 tetramer detection probability approaching 95%. In HEK293 cells, RyR2 expression was dense, with some clusters containing several hundred RyR2s. Ventricular myocytes from mice contained large clusters containing many tens of close-packed RyR2s, resolving apparent discrepancies between electron microscopy and previous super-resolution microscopy data. The methodology developed here reveals the full 3D morphological complexity of RyR2 channels and is applicable to other multi-subunit complexes in a variety of cell types.”